# Relative subsystems and quantum reference frame transformations
Esteban Castro-Ruiz [1,2,3,4] ✉ & Ognyan Oreshkov[1]

Recently there has been much effort in developing a quantum generalisation of reference frame transformations. Despite important progress, a complete understanding of their principles is still lacking. Here we derive quantum reference frame transformations for a broad range of symmetry groups from first principles, using only standard quantum theory. Our framework, naturally based on incoherent rather than coherent group averaging, yields reversible transformations that only depend on the reference frames and system of interest. We find more general transformations than those studied so far, which are valid only in a restricted subspace. Our framework contains additional degrees of freedom in the form of an "extra particle", which carries information about the quantum features of reference frame states. We study the centrally extended Galilei group specifically, highlighting key differences from previous proposals.

Transformations between reference frames play a crucial role in physics. In practice, reference frames are realised by physical systems, which are standardly treated as classical. However, assuming that every physical system is ultimately quantum, it is interesting to ask how a theory of transformations with respect to quantum reference frames (QRFs) would look like, and what implications it would have for our description of the physical world.

The study of QRFs is broad in scope. Seminal works have studied the connection between QRFs and superselection rules[1–5], the study of quantum mechanics with respect to finite-mass QRFs[6–9], quantum tasks and operations under symmetry constraints[3,4,10–21], QRFs as resources of asymmetry[22–29], and QRFs as a means to define physical observables in quantum gravity[30–35].

Recently, attention has turned towards understanding how to change between QRF perspectives, giving rise to formalisms for quantum reference frame transformations[36–54]. Given the description of a physical process with respect to a QRF A, how do we obtain the description from the point of view of QRF B? A precise formulation and answer to this question has the potential to generalise the notion of symmetry and covariance[37,38,40,41,48], with important consequences such as the relativity of entanglement and superposition[37], and the (closely related) relativity of subsystems[52]. It can also provide an operational understanding of spin for relativistic particles[46,47], contribute to understanding the physics of gravitating quantum systems[49,53,55], and to quantum extensions of the general relativistic equivalence principle[55–57].

Despite the important progress done in this line of research, it is safe to say that the principles and operational interpretation of "jumping" from one quantum reference frame to another are not yet fully understood. In particular, as we argue below, previous proposals seem to inevitably encounter the property that reversible transformations between the descriptions relative to different arbitrary QRFs are in general obtained only when these descriptions include the whole rest of the universe. This "non-locality" of the prescriptions is unsettling from a conceptual point of view as it goes contrary to the intuition that predictions concerning local systems should require only local data, raising the question of whether a local approach could be developed.

Here, we derive reversible transformation rules between any two QRFs A and B that only depend on these QRFs and the system S they are used to describe. Our framework holds for unimodular groups, which covers a vast set of symmetries of physical interest. However, we expect that the main principles could be appropriately adopted to even more general groups. Starting form an external observer who uses standard quantum mechanics to describe all internal QRFs and systems, our formulation differs from the purely "internal" approach of[37]. However, both approaches agree when restricted to the fully invariant subspace of pure states. That is, the subspace of pure states $|\psi\rangle$ such that $U(g)|\psi\rangle = |\psi\rangle$ for all $g \in G$, where $U$ is the global action of $G$ on the total Hilbert space. This is precisely the relevant subspace for the "perspective neutral" framework for QRF transformations[38], which obtains the same transformations of[37] for the translation group. In this case, restricting to the trivial subspace means restricting to global states with vanishing total momentum.

Our approach is less restrictive. On purely operational grounds, observers who lack access to the external reference frame are constrained to density operators $\rho$ that are invariant under the action of $G$. This is a weaker requirement than demanding invariance of state vectors $|\psi\rangle$ under $G$.

[1]QuIC, Ecole polytechnique de Bruxelles, C.P. 165, Université libre de Bruxelles, Brussels, Belgium. [2]Institute for Quantum Optics and Quantum Information (IQOQI), Austrian Academy of Sciences, Vienna, Austria. [3]Institute for Theoretical Physics, ETH Zurich, Zurich, Switzerland. [4]Université Paris-Saclay, Inria, CNRS, LMF, Gif-sur-Yvette, France. ✉e-mail: Esteban.CastroRuiz@oeaw.ac.at

Therefore, in this paper we take the view that restrictions purely based on symmetry should be implemented as

$$U(g)\rho U(g)^{\dagger} = \rho \qquad (1)$$

rather than

$$U(g)|\psi\rangle = |\psi\rangle. \qquad (2)$$

This distinction is important, and in this paper we argue in favour of the former option. To illustrate the difference, consider for example the case of the translation group. In our framework, one does not need to specify the value of the total momentum, even less to demand that its value is zero. In general, group theoretic terms, our formalism does not need to specify the value of the total charge, a global invariant quantity, and QRF perspectives are defined locally. The QRF transformation rules that we obtain are therefore different than the ones found in previous works. They are, however, consistent with them provided that the total charge vanishes, a fact that can be checked "internally", as the total charge is an invariant observable.

Essential to our framework is the algebra of an "extra particle," which emerges as a consequence of the invariant degrees of freedom of the reference frame. We argue that the extra particle should be included in the relative description of quantum systems in a standard way. The reason why its importance has not been noticed so far is that we normally deal with sharply-defined, classical reference frames, for which, as we show, the extra particle is always in a maximally mixed state. However, when considering general QRFs, the extra particle should be included, because it is essential for obtaining reversible QRF transformations. As an illustration of the physical meaning of our framework, we analyse quantum reference frame transformations with respect to the (centrally extended) Galilei group.

In the following we argue, via a thought experiment, that existing approaches to QRF transformations are not satisfactory when it comes to adding extra systems to our description of an experiment. The situation we consider is a modification of the so-called "paradox of the third particle," first introduced in ref. 7. (For a comparison between the solution to the paradox offered in ref. 7 and the one offered here, see Subsection Comparison with other frameworks.

Consider a reference frame for spatial translation in nonrelativistic physics. Classically, this is equivalent to a point-like particle (e.g., the centre of mass of a body) that occupies a certain position in space. Since every such particle is ultimately a quantum system, it could in principle also exist in a state that is a quantum superposition of largely different spatial positions. One of the questions that the theory of QRFs is concerned with is how physical systems would be described if one uses a reference frame in such a quantum superposition, and what the transformation rules relating the descriptions relative to different QRFs are.

Imagine that we start from a reference frame A that is well localised in space from the point of view of an observer E. (Ignoring special-relativistic and gravitational considerations, the uncertainty in the position of a particle can in principle be made arbitrarily small at a given instant.) Imagine that we describe two more particles, B and S, each in a pure state, where S is also well localised, say at position $\vec{r}_{S|A}$, relative to A. How should we describe the state of the system S if we use B instead of A as a reference frame for position in space?

If B is well localised itself, say at position $\vec{r}_{B|A}$ relative to A, we are effectively in a classical situation and the answer is given by a classical coordinate transformation: relative to B, we would see S at position $\vec{r}_{S|B} = \vec{r}_{S|A} - \vec{r}_{B|A}$. But what if B is in a quantum superposition of different positions? Since the location of B relative to A is uncertain and A is at a fixed distance from S, the position of S relative to B is uncertain too. But if both A and S are described jointly relative to B, they have to be correlated in the position basis as they are a fixed distance from each other (and the distance is invariant under changing the origin of the coordinate system). This means that S cannot be in a pure state relative to B, even though it is in a pure state relative to A. This shows that the descriptions of S relative to the two reference frames A and B cannot be related by a unitary transformation.

One can propose a potential solution to this problem using the transformation found in ref. 37. There, one obtains reversible transformations between QRFs by including each QRF in the other's description. In this case, the state of AS relative to B can be pure and unitarily related to the state of BS relative to A, without contradicting the expected correlations between A and S in the perspective of B.

However, imagine that in addition to the described particles, we extend the systems under consideration, adding another particle, S′, localised at a fixed position relative to A. Furthermore, we assume that S′ is sufficiently far away from the rest of the systems, so that it cannot influence them in any way. In operational formulations of quantum theory, one is always allowed to add extra systems to the description, so that adding S′ in the perspective of A seems to be innocuous. However, following the same argument as before, the state of AS relative to B could not be pure, since the positions of A and S relative to B must be correlated with the position of S′ relative to B.

Therefore, it seems that the QRF transformation rule of [37] leads to a conflict: not including the system S′ (from the perspective of A) leads to a pure state of AS relative to B, whereas including it leads to a mixed state. This ambiguity entails not only different descriptions of the same physical situation – ultimately a matter of taste—but rather leads to conflicting physical predictions: one can always distinguish a pure state from a mixed one by performing suitable measurements.

A possible answer is that we obtained a contradiction because we failed to include particle S′ in the former analysis. The system S, which for simplicity we assumed to be a single particle here, must in principle contain all particles that are not in translationally invariant states relative to A. Only then are these unitary transformation rules supposed to hold. Indeed, as put forward by the perspective-neural approach[38], and as we will see again here, the unitary transformation rules[37] for jumping between different QRFs for the translation group can be derived assuming that the total system ABS has a vanishing total momentum. A vanishing total momentum can be shown to guarantee, in particular, that, relative to A, there are no systems outside of BS in translationally non-invariant states. This forces the system S′ to be in a state of vanishing momentum, thereby avoiding the paradox.

How shall we interpret physically the condition of zero total momentum? We distinguish two possibilities: (1) The condition of zero total momentum is a constraint, implying a redundancy in our description. This interpretation, however, rejects the possibility of extending our description to other degrees of freedom transforming nontrivially under translations. In a cosmological context, this condition could be naturally justified from a global Dirac constraint on the Hilbert space of the full universe[42]. However, such a constraint is in general only supposed to hold on all physical systems and need not hold for arbitrary subsystems of the universe. Even in field theories like general relativity, where the momentum constraint is local, meaning that the total momentum vanishes at each point in space, there could be different separations of the fields into subsystems, such that the constraint holds for the full system but not for the subsystems. Moreover, such a constraint would arise from the quantisation of a full dynamical theory, given by a specific (translation-invariant) Lagrangian, and not from symmetry considerations alone. Indeed, different Lagrangians with the same symmetry (up to a total derivative) could lead to different constraints (see Methods: Lagrangians for translation gauge symmetry for an example). (2) Alternatively, the system ABS must be explicitly assumed to have a total momentum zero with respect to some external observer E. This, however, is a rather restricted scenario for reasonably confined systems, which in practice cannot capture even the case of localised reference frames such as those that we use in everyday situations.

A natural question then is whether it is at all possible to formulate reversible QRF transformation rules that apply to arbitrary subsystems. As we show in this paper, the answer is positive. Our key insight is that in order to obtain such reversible transformations, we must define the perspective of each frame as containing all invariant degrees of freedom of the reference frame and system of interest, which is a strictly larger set than the set of degrees of freedom describing the system of interest relative to the frame. The end result is a framework that transforms the invariant, operationally-defined description of one observer, who can only perform invariant

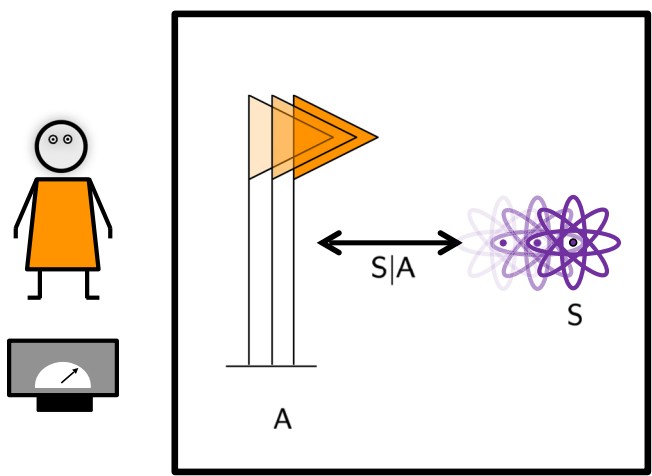

**Fig. 1 | One-party setup.** An observer, Alice, has only access to the degrees of freedom that are invariant under the action of the group $G$. The latter is defined relative to some external observer. As we show in Subsection Relative subsystems, the invariant degrees of freedom are independent of any external observer or reference frame. These invariant degrees of freedom include the system S (in purple) relative to the reference frame A (in orange). They are described by a set of operators forming an algebra, called S|A. Alice's apparatuses, by means of which these degrees of freedom are accessed, are not part of the quantum system under consideration. They lie on the "other side" of Heisenberg's cut.

measurements on a subsystem of a potentially larger system, to the invariant, operationally-defined description of another observer with the same restrictions. Note that this operational interpretation differs from that of the perspective-neutral framework, presented in ref. 40, where a quantum reference frame transformation is a change of mathematical description of a state *before* imposing symmetry invariance.

## Results and discussion
### Modelling a quantum reference frame

Let us now introduce the basic ingredients of our framework. In particular, we define the notion of quantum reference frame that we will use throughout this work.

Consider the one-party setup of Fig. 1. An observer, Alice, possesses a reference frame A, associated with the symmetry group $G$. She uses it to perform quantum operations on a system S, which transforms under some unitary representation of $G$. We treat both A and S quantum mechanically. To do this, we imagine an external observer, Eve, with a reference frame E, who has full access to both systems. Eve assigns a Hilbert space to the composite system

$$\mathcal{H}_{AS|E} = \mathcal{H}_{A|E} \otimes \mathcal{H}_{S|E}. \tag{3}$$

The reason for the notation |E in Eq. (3) is that the quantum mechanical description of A and S is defined with respect to the reference frame of Eve. In the remaining of this section, we will omit this label, as we will be concerned with Eve's description only. However, in Subsection Relative subsystems, this point will be important and we shall introduce the notation again to distinguish it from the "internal" perspective of Alice, who has only access to operators that are invariant under the action of $G$. Eventually, we will do away with the external observer by considering only operators living in the invariant subspace. At this level of description, Eve regards the degrees of freedom of Alice's measurement apparatus (and Alice herself) as implicit. They lie on the "other side" of Heisenberg's cut. If desired, the cut can be moved to include such degrees of freedom explicitly.

To make contact with the standard situation in quantum mechanics, where reference frames are assumed to be classical and are treated implicitly, we assume that the QRF A is perfect. That is, it can be prepared in a basis of states that break the symmetry of $G$ maximally[3]. Therefore, the Hilbert space of A, $\mathcal{H}_A$, is the span of a fully distinguishable basis of "classical" states

labeled by group elements, $|g\rangle_A$. Because basis states are fully distinguishable, we have $\langle g|g'\rangle = \delta(g^{-1}g')$. Here, $\delta(g)$ denotes the Dirac delta distribution for continuous groups, where the group identity element $e$ plays the role of the real number 0, or the (single-argument) Kronecker delta for discrete groups. Thus, $\mathcal{H}_A$ consists of square-integrable functions on $G$ with respect to the invariant measure d$g$. (In this work, we consider only unimodular groups, that is, groups for which the left-invariant and the right-invariant measure are the same.) $\mathcal{H}_A$ carries the left- and right-regular representations of $G$. The left-regular representation, $L_A$, acts as

$$L_A(g)|g'\rangle_A = |gg'\rangle_A, \tag{4}$$

for all $g$ and $g'$ in $G$. The right-regular representation, $R_A$, acts as $R_A(g)|g'\rangle_A = |g'g^{-1}\rangle_A$ or all $g$ and $g'$ in $G$. Both $L_A$ and $R_A$ are unitary representations. The only assumption we make on S is that it transforms under a unitary representation, $U_S$, of $G$. Mathematically, this setup closely resembles that of ref. 4, where the regular representation is used as a token in a quantum communication scheme.

The regular representation is highly reducible—it contains all irreducible representations (irreps) of the group. We can write the basis states $|g\rangle_A$ as[2]

$$|g\rangle_A = \int dq dx dy \sqrt{\frac{\dim(q)}{|G|}} D_{xy}^{(q)}(g)|q; x, y\rangle_A, \tag{5}$$

where $q$ is the "charge" labelling a specific irrep. For compact groups, $\dim(q)$ denotes the dimension of the irrep labeled by $q$, and $|G|$ denotes the order of $G$. The complex numbers $D_{xy}^{(q)}(g)$ are matrix elements of the irrep $q$ for $g \in G$. The left-regular representation $L_A(g)$ acts on the "colour" degrees of freedom, labeled by $x$, whereas the right-regular representation $R_A(g)$ acts on the "flavour" or multiplicity degrees of freedom, labeled by $y$[2]. For the regular representation, the dimension of the multiplicity degrees of freedom for a given irrep $q$ equals the dimension of $q$.

Although Eq. (5) is written under the assumption that both $\dim(q)$ and $|G|$ are finite, a similar equation holds more generally, not only for compact groups. For example, for translations, Eq. (5) reduces to the well-known Fourier transform relation between position eigenvectors $|x\rangle$ and momentum eigenvectors $|p\rangle$: $|x\rangle = (1/\sqrt{2\pi}) \int dp \exp(-ipx)|p\rangle$. Strictly speaking, the vectors $|x\rangle$, more generally $|g\rangle$, are not elements of the Hilbert space. However, we will work with them as is customary in the physics literature. As we will see in Subsection Centrally extended Galilei group, Eq. (5) will be useful in the case of the centrally extended Galilei group, where the quotient $\dim(q)/|G|$ is replaced by the mass parameter, $m$, labeling the irrep.

For an example in the case of compact groups, suppose $G$ is the rotation group $SU(2)$. In this case, $q$ corresponds to the total angular momentum, and the integral with respect to $q$ is replaced by a sum that runs over all values of total angular momentum, or equivalently, all irreps of $SU(2)$. As the labels $x$ and $y$ are discrete, the integral in Eq. (5) is also replaced by a sum running over all possible projections for a given irrep. From Eve's point of view, $G$ acts physically on the colour degrees of freedom of A, leaving the multiplicity degrees of freedom untouched. For $SU(2)$, the action of $G$ corresponds to physically rotating the reference frame A. In this case, the label $x$ corresponds to all the projections of the angular momentum along a specific axis, say $\hat{z}$.

The previous discussion implies that $\mathcal{H}_A$ has the following associated decomposition:

$$\mathcal{H}_A = \bigoplus_q \mathcal{H}_{A_L}^{(q)} \otimes \mathcal{H}_{A_R}^{(q)}, \tag{6}$$

where the direct sum runs over all possible values of the charge $q$. The charge could take discrete or continuous values, where in the latter case the states pertaining to the subspaces labeled by $q$ need to be properly normalised as elements of the full Hilbert space. For the time being we will ignore this technicality, and revisit it again in Subsection Centrally extended Galilei group and Methods: Basis vectors for centrally extended Galilei group.

The left ($\mathcal{H}_{A_L}^{(q)}$) and right ($\mathcal{H}_{A_R}^{(q)}$) tensor factors in each subspace labeled by the charge correspond, respectively, to the colour and flavour degrees of freedom of A. With respect to this decomposition, the left-regular representation has the form $L_A(g) = \bigoplus_q D_{A_L}^{(q)}(g) \otimes \mathbb{1}_{A_R}^{(q)}$, where $D_{A_L}^{(q)}(g)$ is an irrep of $G$ corresponding to the charge $q$. Similarly, the right-regular representation has the form $R_A(g) = \bigoplus_q \mathbb{1}_{A_L}^{(q)} \otimes D_{A_R}^{(q*)}(g)$, where $D_{A_R}^{(q*)}$ denotes the conjugate representation corresponding to the charge $q$. Given a choice of basis as defined in Eq. (5), $D_{A_R}^{(q)*}$ is obtained by complex-conjugating the matrix elements of $D_{R_A}^{(q)}$.

In general, a Hilbert space decomposing as a direct sum of tensor products, like in Eq. (6), is said to decompose into subsystems[58,59]. Here, we will use a slightly more general terminology, associating a subsystem with a subalgebra of operators[60,61]. In particular, we will speak about the left subsystem, which is associated with the subalgebra of operators of the form $T_L = \bigoplus_q T_{A_L}^{(q)} \otimes \mathbb{1}_{A_R}^{(q)}$, and about the right subsystem, which is the commutant of the left, and consists of operators of the form $T_R = \bigoplus_q \mathbb{1}_{A_L}^{(q)} \otimes T_{A_R}^{(q)}$. A given (type-I von Neumann) subalgebra (equivalently, its commutant) always induces a decomposition of the Hilbert space of the form (6)[58,59]. Note that the basis vectors $|g\rangle_A$ generally involve nontrivial superpositions of vectors belonging to the subspaces corresponding to different charges.

What is the physical realisation of an ideal quantum reference frame as defined above? The answer generally depends on the group. In Subsection Centrally extended Galilei group, we will discuss reference frames for the centrally extended Galilei group. We will show that for this group a reference frame is physically equivalent to two particles—one that serves as a reference for position and the other one as a reference for velocity.

### Relative subsystems

Here we construct the description of the setup in Fig. 1 from Alice's reference frame. First, we find the subsystem of the full A and S system that Alice has access to. Afterwards, we construct a map form the Hilbert space associated to the external observer, Eve (see Subsection Modelling a quantum reference frame), to a Hilbert space with a tensor product structure that is natural from the point of view A. This map entails a refactorisation of the Hilbert space, which can be interpreted as "jumping" into Alice's reference frame. We study how the representation of the invariant subsystem changes under this refactorisation. We find that the full invariant subsystem is larger than the algebra of relative observables between the system and frame. It contains an extra subsystem, which we call the "extra particle," due to its physical realisation in the case of the Galilei group, discussed in Subsection Centrally extended Galilei group.

**The invariant subsystem.** From Eve's perspective, the Hilbert space of A and S factorises as $\mathcal{H}_{AS|E} = \mathcal{H}_{A|E} \otimes \mathcal{H}_{S|E}$. We call this tensor product factorisation the standard partition. In the standard partition, $G$ acts transversally on operators $T$, as $T \mapsto L_A(g) \otimes U_S(g) T L_A^\dagger(g) \otimes U_S^\dagger(g)$, for $g \in G$. Throughout, we assume that $G$ is a unimodular group and that the Hilbert space on which it acts is separable. Unless otherwise stated, all operators are assumed bounded.

What are the degrees of freedom that Alice has access to, and how would she describe them? By assumption, Alice has no access to the external reference frame E. Therefore, she has only access to the $G$-invariant degrees of freedom of the AS system. That is, operators on $\mathcal{H}_{AS|E}$ that are invariant under the transversal action of $G$: $T = L_A(g) \otimes U_S(g) T L_A^\dagger(g) \otimes U_S^\dagger(g)$, for all $g \in G$. Note that she has access to all these degrees of freedom. This fact can be derived from the description of relative operators given in Eq. (8): if an observer loses access to the reference frame relative to which their description of the system is given, they would still be able to make sense of the subset of relative operators that are localised entirely on the system, and these are exactly the set of invariant operators on the system. This argument justifies our use of an incoherent group averaging approach to QRFs.

Therefore, we see that symmetry considerations alone do not imply the coherent group averaging approach. Indeed, when we use coherent twirling instead of incoherent, we do more than implementing a symmetry resulting

from the lack of a reference frame—we impose a charge sector. Moreover, as we show in Methods: Group action in A's decomposition, this result is independent of the external reference frame that is assumed in the derivation, which justifies lifting it to a general principle that holds even when an external reference frame does not exist physically.

The set of all bounded $G$-invariant operators forms an algebra, which we call the invariant subsystem. We assume that Alice has access to all of these (and only these) operators.

Note that any unitary representation of a locally compact group $G$ on a separable Hilbert space $\mathcal{H}$ induces an analogous decomposition to that in Eq. (6), $\mathcal{H} = \bigoplus_q \mathcal{J}^{(q)} \otimes \mathcal{K}^{(q)}$, such that $G$ acts irreducibly on each $\mathcal{J}^{(q)}$ and trivially on each $\mathcal{K}^{(q)}$. In general, the labels $q$ need not go over all possible irreps, like in the case of the regular representation, and the Hilbert spaces $\mathcal{K}^{(q)}$ need not be of the same dimension as $\mathcal{J}^{(q)}$. This decomposition is a consequence of the fact that a generally reducible representation splits into a direct sum of irreps, some of which might have nontrivial multiplicities. By Schur's lemma, all invariant operators are proportional to the identity on $\mathcal{J}^{(q)}$ for all $q$ and are possibly nontrivial on the multiplicity factors $\mathcal{K}^{(q)}$. These operators form the invariant algebra, or the invariant subsystem. Its commutant—which is the algebra with trivial action on the multiplicity factors $\mathcal{K}^{(q)}$—is what we call the gauge subsystem. For example, in the case of the Galilei group for a system of particles, the gauge subsystem corresponds to the centre of mass degrees of freedom[7-9].

In our case, any operator on the gauge subsystem is physically irrelevant for Alice—it is redundant. This redundancy can be removed by aplying a superoperataor projector $\mathcal{T}_{AS}$ that projects the algebra of operators over the Hilbert space onto the invariant algebra. In the case of compact groups, this projector is given by the G-twirl[3],

$$\mathcal{T}_{AS} = \int dg\, L_A(g) \otimes U_S(g) \cdot L_A^\dagger(g) \otimes U_S^\dagger(g). \tag{7}$$

As shown in ref. 3, this operation is equivalent to first projecting the operator into a block-diagonal form over the charge sectors (i.e., killing off-diagonal elements between subspaces corresponding to different charges), followed by applying fully depolarising channels in the left tensor factors. In the standard partition, the space of physically relevant (bounded) operators from the point of view of Alice, denoted by $\mathcal{B}_{inv}(\mathcal{H}_{AS|E})$, is defined by those operators which are invariant under the $G$-twirl, $T_{inv} = \mathcal{T}_{AS}[T_{inv}]$. $\mathcal{B}_{inv}(\mathcal{H}_{AS|E})$ is a proper subspace of the vector space of operators on $\mathcal{H}_{AS|E}$, called $\mathcal{L}(\mathcal{H}_{AS|E})$.

Note that $\mathcal{B}_{inv}(\mathcal{H}_{AS|E})$ is independent of Eve's external reference frame, E, with respect to which the systems A and S, and the action of $G$ were defined. More precisely, as we show in Methods: Independence of external frame, the invariant algebra of a given system (in this case AS) is the largest common subalgebra of the "relative algebra" (to be defined precisely shortly) AS|E for all conceivable external reference frames E. The invariant algebra $\mathcal{B}_{inv}(\mathcal{H}_{AS|E})$ can thus be regarded as meaningful on its own. We can imagine external reference frames being "out there" or not; our framework is agnostic to their existence.

Let us now turn to Alice's perspective on S. Imagine that Alice describes an operator $T$ acting on the system from her point of view. What would be the corresponding operator in the standard tensor product decomposition? We denote the operator $T_S$ on S relative to A by $T_{S|A}$. All operations on S from Alice's viewpoint correspond to elements of the algebra of system S relative to reference frame A, denoted S|A. In the standard partition, elements $T_{S|A} \in$ S|A are of the form[3,4,13,14]

$$T_{S|A} = \int dg\, |g\rangle \langle g|_A \otimes U_S(g) T_S U_S^\dagger(g), \tag{8}$$

where $T_S$ is an operator on $\mathcal{H}_{S|E}$. Seen as an abstract mathematical object, S|A is independent of the choice of tensor product decomposition: as we will see below, it can have different representations, which are natural to the viewpoint of different reference frames. A rough analogy is that of a point or

a tangent vector to a manifold, which can be represented in different coordinate systems, which are natural from the viewpoint of different observers.

Note that S|A is not the full algebra of $G$-invariant operators. This is because the reference frame A lives in a Hilbert space that carries the regular representation of $G$, which is reducible (see Eq. (6)). As such, it has multiplicity subspaces that are invariant under the action of $G$[2,3]. The multiplicity degrees of freedom are invariant under the transversal action of $G$, as this action is defined in terms of the left-regular representation. As a consequence, any operator $T_R$ on $\mathcal{H}_{AS|E}$ of the form

$$T_R = \bigoplus_q \mathbb{1}_{A_L}^{(q)} \otimes T_{A_R}^{(q)} \otimes \mathbb{1}_S \qquad (9)$$

is $G$-invariant. Here, the first tensor factor denotes the subsystem of A where $L_A(g)$ acts, the second denotes the subsystem of A where $R_A(g)$ acts (see Eq. (6)) and the third one denotes S' degrees of freedom (all in Eve's standard partition). Note that operators of the form (9) generally overlap with S|A, but do not belong to it. Therefore, the full invariant system is strictly larger than S|A. This fact will be important in what follows, as we shall introduce an "extra particle" belonging to the full invariant system.

**Change of preferred tensor product factorisation**. We now construct a representation of the invariant subsystem that captures Alice's perspective in a natural way. Namely, a representation that (i) contains only degrees of freedom accessible to Alice (i.e., it is gauge-free), (ii) contains S|A as an explicit tensor factor. We call this representation "Alice's perspective." This term is motivated by the conventional treatment of subsystems in quantum mechanics, where each subsystem has a tensor factor of its own (more generally, as noted in Subsection Modelling a quantum reference frame, a subsystem is associated with a subalgebra). Thus, when Alice refers to "the system," she is implicitly referring to the system relative to her reference frame. Alice's perspective makes this fact explicit. Moreover, it is justified from an operational perspective (see for example ref. 62), where Hilbert space operators represent experimental procedures defined with respect to laboratory instruments – Alice's reference frame, A, in this case.

The first step is to note that there exists an alternative factorisation of $\mathcal{H}_{AS|E}$ that is induced by the algebra S|A and its commutant, C:

$$\mathcal{H}_{AS|E} \cong \mathcal{H}_C \otimes \mathcal{H}_{S|A} =: \mathcal{H}_{C,S|A}. \qquad (10)$$

To see that this is the case, let us construct explicitly a Hilbert space isomorphism. First, define

$$F_{E \to A}|g\rangle_{A|E} \otimes |\alpha\rangle_{S|E} = |g\rangle_C \otimes |\alpha\rangle_{S|A}, \qquad (11)$$

where $|\alpha\rangle_{S|E}$ and $|\alpha\rangle_{S|A}$ are fixed yet arbitrary bases of $\mathcal{H}_{S|E}$ and $\mathcal{H}_{S|A}$, respectively. The isomorphism can then be written as a map $V_{E \to A} : \mathcal{H}_{AS|E} \longrightarrow \mathcal{H}_{C,S|A}$, defined by $V_{E \to A} = F_{E \to A} \circ U_S^\dagger(\hat{g}_A)$, where

$$U_S^\dagger(\hat{g}_A) = \int dg\, |g\rangle \langle g|_A \otimes U_S^\dagger(g). \qquad (12)$$

As a consequence of the orthogonality of the vectors $|g\rangle_A$, $U_S^\dagger(\hat{g}_A)$ is a unitary operator on $\mathcal{H}_{AS|E}$. It then follows that $\mathcal{H}_C$ carries the left- and right-regular representations of $G$, and $\mathcal{H}_{S|A}$ carries a representation $U_{S|A}$ of $G$ which is isomorphic to $U_S$. (A transformation of the form of Eq. (12) is called a "trivialisation map" or a "disentangler" in ref. 38 and ref. 52.)

A straightforward calculation shows that the super-operator $\mathcal{V}_{E \to A} = V_{E \to A} \cdot V_{E \to A}^\dagger$ maps the representation of S|E in $\mathcal{H}_{AS|E}$ to the tensor factor $\mathcal{H}_{S|A}$,

$$\mathcal{V}_{E \to A}\left[\int dg\, |g\rangle\langle g|_A \otimes U_S(g) T_S U_S^\dagger(g)\right] = \mathbb{1}_C \otimes T_{S|A}. \qquad (13)$$

where $\langle \alpha|_S T_S|\beta\rangle_S = \langle \alpha|_{S|A} T_{S|A}|\beta\rangle_{S|A}$.

Note that, from Alice's perspective, operators on $\mathcal{H}_{C,S|A}$ are not redundancy-free. This is because we have not projected out the gauge subsystem as in Eq. (7). To do so, we use that $\mathcal{V}_{E \to A}$ maps the gauge subsystem to the left-regular representation of $\mathcal{H}_C$:

$$\mathcal{V}_{E \to A}[L_A(g) \otimes U_S(g)] = L_C(g) \otimes \mathbb{1}_{S|A}. \qquad (14)$$

We prove Eq. (14) in Methods: Group action in A's decomposition. Therefore, we can equivalently eliminate the gauge degrees of freedom from any operator by projecting it onto the operator subspace that is invariant under the action of the left-regular representation in $\mathcal{H}_C$. Let $\mathcal{T}_C = \mathcal{V}_{E \to A} \circ \mathcal{T}_{AS} \circ \mathcal{V}_{E \to A}^\dagger$. Using Eq. (14), it is straightforward to verify that this is a superoperator projector on the invarinat subspace with respect to the left-regular representation in $\mathcal{H}_C$. The full procedure of refactorising the Hilbert space and eliminating the redundancy is captured by the map $\mathcal{E}_A = \mathcal{T}_C \circ \mathcal{V}_{E \to A} = \mathcal{V}_{E \to A} \circ \mathcal{T}_{AS}$. In Methods: Explicit form of the $\mathcal{E}_A$ transformation, we obtain a useful expression for this transformation in the case of compact groups. This means that removing the redundancy and changing the factorisation commute in a natural way.

Following the reasoning leading to Eq. (6) and the discussion below it, we see that all operators in $\mathcal{B}_{inv}(\mathcal{H}_{C,S|E})$ are of the form

$$T_{inv} = \bigoplus_q \mathbb{1}_{C_L}^{(q)} \otimes T_{C_R,S|A}^{(q)}, \qquad (15)$$

where $\mathcal{T}_{C_R,S|A}^{(q)}$ is an operator on $\mathcal{H}_{C_R}^{(q)} \otimes \mathcal{H}_{S|A}$, with a notation analogous to that of Eq. (6). Clearly, the identity operators $\mathbb{1}_{C_L}^{(q)}$ are not physically meaningful for Alice, as she cannot access the gauge subsystem. For this reason, we could define Alice's perspective by projecting Eq. (15) on each charge sector $q$ and then tracing out the corresponding $\mathcal{H}_{C_L}^{(q)}$ Hilbert space. However, we will keep the operators $\mathbb{1}_{C_L}^{(q)}$ as in Eq. (15) for mathematical convenience, as will be clear in Subsection Quantum reference frame transformations.

To summarise, in the perspective of A, the full Hilbert space is associated with the following decomposition:

$$\mathcal{H}_{C,S|A} = \left(\bigoplus_q \mathcal{H}_{C_L}^q \otimes \mathcal{H}_{C_R}^q\right) \otimes \mathcal{H}_{S|A}, \qquad (16)$$

where the left subsystem of C contains the gauge degrees of freedom.

**The extra particle**. What is the physical meaning of the right-regular subsystem of C? To answer this question, consider a general operator on C, $\int dg'dg T(g',g)|g'\rangle \langle g|_C \otimes \mathbb{1}_{S|A}$, and act on it with $\mathcal{T}_C$. The result is

$$T_{inv} = \int dg'\, dg T(g',g) R_C^\dagger(g') R_C(g) \otimes \mathbb{1}_{S|A}. \qquad (17)$$

$T_{inv}$ is $G$-invariant, and therefore represents a physically meaningful operator, expressed in Alice's perspective. We call the set of these operators the algebra $\overline{S|A}$. It is the complement of S|A in the full invariant subsystem, $\mathcal{B}_{inv}(\mathcal{H}_{C,S|A})$, in the sense that its tensor product with S|A gives the full invariant subsystem, $\mathcal{B}_{inv}(\mathcal{H}_{C,S|A}) = S|A \otimes \overline{S|A}$.

In the standard partition, $\overline{S|A}$ corresponds to a subsystem which is non-trivial in both the right-regular representation and the system, as can be seen by applying the inverse of Eq. (12) to a general operator on $\overline{S|A}$. Explicitly, in the standard partition, $\overline{S|A}$ consists of operators of the form

$$T_{\overline{S|A}} = \int dg'\, dg|g'\rangle \langle g'|T_{A|E}^R|g\rangle \langle g|_{A|E} \otimes U_{S|E}(g') U_{S|E}^\dagger(g), \qquad (18)$$

where $T_{A|E}^R$ is left-invariant.

We call the algebra $\overline{S|A}$ the "extra particle," because it formally satisfies (in a single mass sector) the algebra of a single particle in the case of the

centrally extended Galilei group, as we show in Subsection Centrally extended Galilei group. As we will see in Subsection Quantum reference frame transformations, $\overline{S|A}$ is essential to the unitarity of quantum reference frame transformations at the level of algebra of observables. For this reason, we argue that, in a fully relative formulation of quantum mechanics, the "extra particle" has to be considered standardly when we refer to a quantum system. In this way, the relative nature of quantum objects with respect to a reference frame, which is normally considered implicit, becomes explicit in our formalism. Moreover, the extra particle is key to making our formalism consistent with the potential existence of an external frame, thus solving the problem presented in the Introduction.

One might wonder under which circumstances the extra particle does not play a significant role and can be considered implicitly. This is the case when the state of the reference frame A in E's factorisation is classical, that is, for states on $\mathcal{H}_{AS}$ of the form $|g\rangle\langle g|_A \otimes \rho_S$ for $g \in G$ and $\rho_S$ a state on $\mathcal{H}_S$, or any convex combination (probabilistic mixture) of such states. Applying $\mathcal{T}_C \circ \mathcal{V}_{E\to A}$ to any such state, we immediately see that the extra particle $\overline{S|A}$ is in the maximally mixed sate and in a tensor product with the state of S|A. In this sense, the extra particle carries information about the "quantumness" of the reference frame state. This "quantumness" is independent of any potential external observer, as $\overline{S|A}$ is part of the invariant subsystem. This implies that our framework distinguishes between a coherent superposition of states related by a "gauge transformation" and a classical mixture thereof (both as defined in Eve's perspective). This is not surprising: these two states are indeed different, and there is no reason to expect that they should coincide in the invariant subsystem. Alice can check this difference even if she does not have access to Eve's reference frame. However, she cannot see the difference by doing operations on S|A alone—the algebra $\overline{S|A}$ is key in this distinction. This fact distinguishes our approach from the perspective-neutral framework for quantum reference frame transformations, in which a coherent superposition and a mixture of states related by a "gauge transformation" are mapped to the same zero-charge state[63]. The reason behind this is that the perspective neutral framework does not consider the information carried by $\overline{S|A}$ and its correlations with S|A.

**Quantum reference frame transformations**

Consider now 2 observers, Alice and Bob, with QRFs A and B, respectively. The total Hilbert space in the standard partition is $\mathcal{H} = \mathcal{H}_A \otimes \mathcal{H}_B \otimes \mathcal{H}_S$ (we omit the explicit reference to Eve's reference frame for simplicity). As before, A and B are perfect reference frames, so $\mathcal{H}_A$ and $\mathcal{H}_B$ each carry the left- and right-regular representation of $G$. $\mathcal{H}_S$ carries an arbitrary unitary representation of $G$. Following the procedure of Subsection Relative subsystems, we can express the invariant subsystem of the joint system ABS in the perspective of Alice. This gives rise to the invariant subalgebra $\mathcal{B}_{inv}(\mathcal{H}_{C,BS|A})$, where $\mathcal{H}_{C,BS|A} = \mathcal{H}_C \otimes \mathcal{H}_{BS|A} = \mathcal{H}_C \otimes \mathcal{H}_{B|A} \otimes \mathcal{H}_{S|A}$, with obvious notation. The space $\mathcal{H}_C$ decomposes into a left- and a right-invariant part. The left-invariant part is the subsystem $\overline{BS|A}$ and the right-invariant part is the gauge subsystem.

An analogous procedure gives rise to Bob's perspective, corresponding to the algebra $\mathcal{B}_{inv}(\mathcal{H}_{D,AS|B})$. As in the case of Alice, $\mathcal{H}_D$ decomposes into a left- and a right-invariant parts, which are the extra particle $\overline{AS|B}$ and the gauge subsystem from B's perspective, respectively. In what follows, we construct a unitary map that relates Alice's and Bob's perspectives. To this end, we note that both perspectives are unitarily related to the standard decomposition (E). Then, to "jump" between the perspective of A and B, we can map the representation of A to that of E and then map the representation of E to that of B. This same logic is used to relate different QRFs in the "perspective neutral" approach[38].

We define a quantum reference frame transformation from Alice to Bob, $\mathcal{S}_{A\to B} : \mathcal{B}_{inv}(\mathcal{H}_{C,BS|A}) \longrightarrow \mathcal{B}_{inv}(\mathcal{H}_{D,AS|B})$, as

$$\mathcal{S}_{A\to B} = \mathcal{V}_{E\to B} \circ \mathcal{V}_{E\to A}^\dagger. \tag{19}$$

Following the same logic as in Subsection Change of preferred tensor product factorisation, we define $\mathcal{V}_{E\to B} = V_{E\to B} \cdot V_{E\to B}^\dagger$ and

$\mathcal{V}_{E\to A}^\dagger = V_{E\to A}^\dagger \cdot V_{E\to A}$. Here, $\mathcal{V}_{E\to B} = F_{E\to B} \circ U_{AS}^\dagger(\hat{g}_B)$ and $\mathcal{V}_{E\to A} = F_{E\to A} \circ U_{BS}^\dagger(\hat{g}_A)$, where

$$U_{BS}(\hat{g}_A) = \int dg\, |g\rangle\langle g|_A \otimes L_B(g) \otimes U_S(g), \tag{20a}$$

$$U_{AS}(\hat{g}_B) = \int dg\, L_A(g) \otimes |g\rangle\langle g|_B \otimes U_S(g). \tag{20b}$$

$F_{E\to A}$ acts as $F_{E\to A}|g\rangle_A|h\rangle_B|\alpha\rangle_S = |g\rangle_C|h\rangle_{B|A}|\alpha\rangle_{S|A}$, and an analogous equation holds for $F_{E\to B}$.

The quantum reference frame transformation of Eq. (19) generalises the one of ref. 37 for Lie groups by including the algebra of the extra particle. To see this, we write Eq. (19) in a similar form to that of ref. 37. Assume that $G$ is a Lie group such that, for any $g \in G$, we can write $U(g) = \exp(-i\lambda_g \cdot X)$, where $\lambda_g$ is a vector of parameters corresponding to $g$ and $X$ is a vector whose components are the generators of the Lie algebra of $G$. Under these conditions, as shown in Methods: Exponential representation of quantum reference frame transformation, we arrive at the following form

$$S_{A\to B} = \mathcal{P}_{A\to B} e^{i\int dg \lambda_g |g\rangle\langle g|_{B|A} \cdot (X_{\overline{BS|A}} + X_{S|A})}, \tag{21}$$

where we have left tensor products with the identity operator implicit. Here, $X_{\overline{BS|A}}$ is the infinitesimal generator acting on the extra particle $\overline{BS|A}$ and $X_{S|A}$ is the infinitesimal generator on the subsystem S|A. The parity-swap operator $\mathcal{P}_{A\to B}$ acts as

$$\mathcal{P}_{A\to B}|g\rangle_{B|A} = |g^{-1}\rangle_{A|B}, \tag{22}$$

with an implicit trivial action on all subsystems other than B|A. A few comments are in order: (1) The transformation of Eq. (21) includes extra degrees of freedom in the form of the extra particle $\overline{BS|A}$; (2) the transformation is block diagonal, with each block corresponding to a different irreducible representation of $G$, labeled by $q$, so the choice $q = 0$ is not necessary and we can focus on any sector for arbitrary $q$; (3) for the special case $q = 0$, the transformation is compatible with that of ref. 37. Consider the translation group as an example. In this case, Eq. (21) reads

$$S_{A\to B} = \mathcal{P}_{A\to B} e^{i\hat{x}_{B|A}\left(\hat{p}_{\overline{BS|A}} + \hat{p}_{S|A}\right)}, \tag{23}$$

which differs form the one of ref. 37 due to the extra term $\hat{p}_{\overline{BS|A}}$. In the case of zero total momentum, this term vanishes and both transformations are equal.

We can now use Eq. (19) to compute the transformation of operators from the perspective of A to that of B. Let us divide the set of operators in the reference frame of A into 3 classes. Class 1 is made of operators of the form $\mathbb{1}_C \otimes \mathbb{1}_{B|A} \otimes T_{S|A}$, i.e., elements of S|A; class 2 is made of operators of the form $\mathbb{1}_C \otimes T_{B|A} \otimes \mathbb{1}_{S|A}$, i.e., elements of B|A. Finally, class 3 is made of operators of the form $T_C^R \otimes \mathbb{1}_{B|A} \otimes \mathbb{1}_{S|A}$, where $T^R$ is left-invariant, i.e., elements of $\overline{BS|A}$. The transformation of each of these 3 classes of operators is computed explicitly in Methods: Transformation of relative subsystems. The result is

$$\mathcal{S}_{A\to B}[\mathbb{1}_C \otimes \mathbb{1}_{B|A} \otimes T_{S|A}] = \int dg\, |g\rangle\langle g|_{A|B} \otimes \mathbb{1}_D \otimes U_{S|B}(g) T_{S|B} U_{S|B}^\dagger(g) \tag{24a}$$

$$\mathcal{S}_{A\to B}[\mathbb{1}_C \otimes T_{B|A} \otimes \mathbb{1}_{S|A}] = \int dh dg\, |h^{-1}\rangle\langle h|T_{A|B}|g\rangle \langle g^{-1}|_{A|B} \otimes R_D(h^{-1}g) \otimes U_{S|B}(h^{-1}g) \tag{24b}$$

$$\mathcal{S}_{A\to B}[T_C^R \otimes \mathbb{1}_{B|A} \otimes \mathbb{1}_{S|A}] = \int dg\, |g\rangle\langle g|_{A|B} \otimes R_D(g) T_D^R R_D^\dagger(g) \otimes \mathbb{1}_{S|B}. \tag{24c}$$

Equation (24) fully characterise the relation between A's natural tensor product factorisation and B's. We thus see that a quantum reference frame is a preferred tensor factorisation of the invariant subsystem. Alice and Bob have 2 such partitions, natural to their relative degrees of freedom. This fact is at the heart of the relativity of entanglement under QRF transformations[37,52]. As we show in Methods: Restriction to the zero-charge sector, in the zero-charge sector Eq. (19) reduces to the QRF transformation found in ref. 39, which is equivalent to that of ref. 37 for the case of translations.

Note that S|A and S|B partially overlap but are not equal. The same is true for the subalgebras BS|A and AS|B. For this reason, we cannot expect these subalgebras to be unitarily related. However, the extra particle comes to the rescue, as it complements each of BS|A and AS|B to the full invariant subsystem. This is why the extra particle is essential for unitarity.

It is worth emphasising the generality of the transformations in Eqs. (24). They do not merely allow us to say how to "jump" between two fixed reference frames, but also how the description from the point of view of one reference frame would change if that reference frame is subjected to an arbitrary active transformation from the perspective of another. For instance, if Alice applies an active unitary transformation on B, $U_{B|A}$, the state of the invariant subsystem in the perspective of Bob would undergo a corresponding passive unitary transformation, whose form can be computed from Eq. (24b) by plugging $U_{B|A}$ in the place of $T_{B|A}$. The transformation seen by Bob would generally spread over the system, Alice's frame, as well as the extra particle, where the latter is again essential for recovering unitarity (see Methods: Extra particle and unitarity). The transformations of Eq. (24) are obviously not restricted to scenarios involving two reference frames, as additional frames can be included in S.

Note that the extra particle $\overline{S|A}$ arising in the description of a given system S by a given frame A overlaps with any relative system, such as B|A, that may be brought into the description relative to A, and in this sense can be said to contain information about the "rest of the world" relative to A (of course, if we consider the extra particle $\overline{SB|A}$, it would be separate from both S|A and B|A). This explains why, if we jump from a classical frame to a frame in a superposition, any system that was previously in a pure non-invariant state would look correlated with any other such system in the universe, yet its purification can be found in its corresponding extra particle without violating the monogamy of entanglement. It is important to stress, however, that even though the extra particle overlaps with additional systems (including other reference frames) relative to the frame in question, it does not contain gauge degrees of freedom as it is fully within the invariant subsystem of the system and frame.

To summarise, our framework decomposes the full invariant subsystem as a network of subsystems, whose "threads" represent the viewpoints of A and B. A QRF transformation is a change from a decomposition which is natural to Alice to a decomposition which is natural to Bob. Figure 2 depicts how each subalgebra in Alice's reference frame commutes or fails to commute with each subalgebra in Bob's partition. The vertical "threads" correspond to Alice's QRF, whereas the horizontal ones correspond to Bob's QRF. More generally, we can imagine multiple reference frames and the corresponding network of relative subalgebras related via analogous principles. A feature of these algebraic relations is that, as commented earlier (see Methods: Independence of external frame, they concern algebras that are independent of external reference frames, yet compatible with any potential external reference frame in the sense that they would automatically embed as subalgebras of the corresponding larger invariant algebra entailed by the existence of such a frame. This unveils a mathematical landscape of nested subalgebras that may represent both actual and potential scenarios.

### Centrally extended Galilei group

In this Subsection, we apply our framework to the case of the centrally extended Galilei group. We start by briefly introducing the Galilei group and its central extension. Then, we compute the algebras S|A and $\overline{S|A}$, and give a physical interpretation of the regular representation as a quantum reference frame. For simplicity, we treat the case of 1 spatial dimension and focus only on spatial translations and boosts, leaving time translations to further

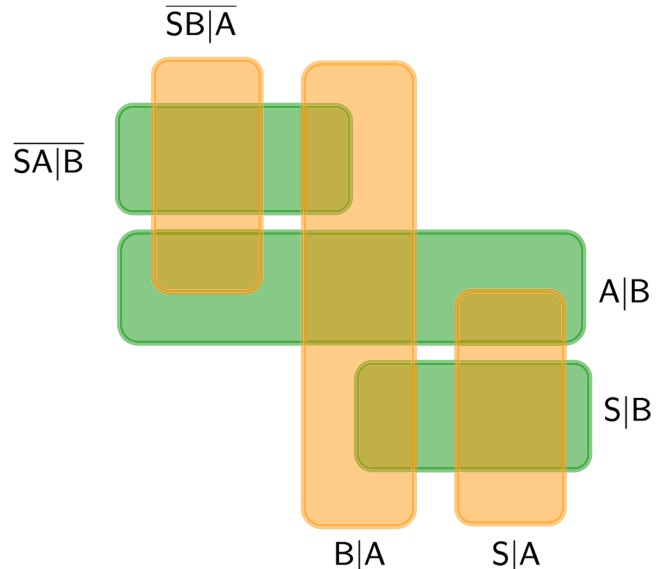

**Fig. 2 | Natural subsystem decompositions.** The full invariant system can be decomposed in a way that is natural to A (vertical, orange "threads") and in a way that is natural to B (horizontal, green threads). A QRF is a preferred factorisation of the invariant system, and a QRF transformation is a change from one preferred factorisation to another. In this illustration, when 2 different subsystems overlap it means that their corresponding operators don't commute in general. In this way, when A refers to "the system," she is actually referring to the subsystem A|B, which overlaps with S|B and A|B from the point of view of B. Note that the inclusion of the subsystems $\overline{SB|A}$ and $\overline{SA|B}$ is essential to find a unitary relation between A's and B's tensor product factorisations.

work. Although our treatment is formal, glossing over normalisation issues and applying our theory to unbounded operators (strictly speaking, it is developed for bounded operators only), we extract the essential physics and obtain compelling insights about the physical realisation of the regular representation as a QRF. It would be interesting to see if our construction can be cast in the rigorous formulation of covariant "screen observables" for the Galilei and Poincaré groups[64]. In addition, the recent developments of ref. 65 might also be helpful in this regard.

**Introducing the group.** In 1 spatial dimension, the Galilei group consist in elements $(a, v)$, labeled by a translation parameter $a \in \mathbb{R}$ and a boost parameter $v \in \mathbb{R}$. Physically, the transformation $(a, v)$ means changing to a reference frame which is displaced in space by a distance $a$ and moving with a constant velocity $v$ with respect to the original reference frame. The composition rule of the Galilei group is $(a', v') \cdot (a, v) = (a' + a, v' + v)$.

Galilean transformations on a quantum particle of mass $m$ are generated by the momentum operator $\hat{p}$ (translations) and by the boost operator $\hat{k} = \hat{p}t - m\hat{x}$ (boosts), where $t$ is the time and $\hat{x}$ is the position operator. The commutation relation of the group is $[\hat{p}, \hat{k}] = im$. The non-commutativity of the Galilean generators in quantum mechanics implies the well known fact that the Galilean group has a projective representation in Hilbert space

$$U^{(m)}(a', v')U^{(m)}(a, v) = e^{i\frac{m}{2}(av' - a'v)}U(a' + a, v' + v), \quad (25)$$

where $U(a, v) = \exp(-i(a\hat{p} + v\hat{k}))$. In order to apply our framework, we consider the central extension of the Galilei group, $\tilde{G}$ (see, for example,[66,67]). $\tilde{G}$ has group elements $(\theta, a, v)$ and group multiplication rule $(\theta', a', v') \cdot (\theta, a, v) = (\theta' + \theta + \varphi(a', v'; a, v), a' + a, v' + v)$, where $\varphi(a', v'; a, v) = (av' - a'v)/2$. For a given mass $m$, we define the (irreducible) representation of $\tilde{G}$ by $\tilde{U}^{(m)}(\theta, a, v) = e^{im\theta}U^{(m)}(a, v)$. It is easy to check that Eq. (25) is an ordinary (i.e., not projective) representation of the centrally extended Galilei group.

The centrally extended Galilei group involves an additional parameter, $\theta$, whose physical meaning as the conjugate variable to a dynamical mass variable has been discussed in the literature[67–71]. Regardless the specific meaning of $\theta$, we do not miss any physics by conceiving a (possibly fictitious) reference frame for it, since the physically accessible projective representations of the Galilei group are naturally recovered in the case of reference frames of fixed mass. What is more, this treatment highlights the interesting possibility of having an explicit QRF for dynamical mass, allowing for coherences between different mass sectors.

**QRFs for the centrally extended Galilei group.** Suppose that A is a QRF carrying the regular representation of $\tilde{G}$. This representation is spanned by vectors of the form

$$|(\theta, a, v)\rangle = \int^{\oplus} dm\, dp\, \sqrt{m}\, (\tilde{U}^{(m)}(\theta, a, v)|m; p\rangle_L) \otimes |m; p\rangle_R, \quad (26)$$

with $\theta, a, v \in \mathbb{R}$[66], and has inner product (see Methods: Basis vectors for centrally extended Galilei group)

$$\langle(\theta', a', v')|(\theta, a, v)\rangle = \delta(\theta' - \theta)\delta(a' - a)\delta(v' - v). \quad (27)$$

Consider a system S carrying an irreducible representation of $\tilde{G}$, labeled by the mass $m_S$. This is equivalent to a particle of mass $m_S$. (While we are considering a single such system, our results apply automatically to a system of multiple particles, where the role of S would be played by the centre of mass.) Using Eq. (8) we can compute the generators of Galilean transformations, $\hat{p}_{S|A}$ and $\hat{k}_{S|A}$, in the standard partition. As shown in Methods: Relative generators for the centrally extended Galilei group, the result is

$$\hat{p}_{S|A} = \mathbb{1}_A \otimes \hat{p}_S - m_S \int d\theta\, da\, dv\, v\, |(\theta, a, v)\rangle \langle(\theta, a, v)|_A \otimes \mathbb{1}_S, \quad (28a)$$

$$\hat{k}_{S|A} = \mathbb{1}_A \otimes \hat{k}_S + m_S \int d\theta\, da\, dv\, a\, |(\theta, a, v)\rangle \langle(\theta, a, v)|_A \otimes \mathbb{1}_S. \quad (28b)$$

Note that the integral terms in Eqs. (28a) and (28b) can be interpreted as velocity and position operators on the Hilbert space of the reference frame. In terms of the decomposition into irreducible representations of the reference frame, the generators read

$$\hat{p}_{S|A} = \mathbb{1}_A \otimes \hat{p}_S - m_S \int^{\oplus} \frac{dm}{m} \left( \hat{p}_{A_L}^{(m)} \otimes \mathbb{1}_{A_R}^{(m*)} - \mathbb{1}_{A_L}^{(m)} \otimes \hat{p}_{A_R}^{(m*)} \right) \otimes \mathbb{1}_S, \quad (29a)$$

$$\hat{k}_{S|A} = \mathbb{1}_A \otimes \hat{k}_S - m_S \int^{\oplus} \frac{dm}{m} \left( \hat{k}_{A_L}^{(m)} \otimes \mathbb{1}_{A_R}^{(m*)} - \mathbb{1}_{A_L}^{(m)} \otimes \hat{k}_{A_R}^{(m*)} \right) \otimes \mathbb{1}_S, \quad (29b)$$

where we have taken $t = 0$ for simplicity. Here, $\mathbb{1}_A = \int^{\oplus} dm\, \mathbb{1}_{A_L}^{(m)} \otimes \mathbb{1}_{A_R}^{(m*)}$, and we have used the same notation as in Subsection Relative subsystems.

We can use Eqs. (29) to compute the algebra of the extra particle in the standard partition (see Methods: Relative generators for the centrally extended Galilei group):

$$\hat{p}_{\overline{S|A}} = \hat{p}_A^R \otimes \mathbb{1}_S + \mathbb{1}_A \otimes \hat{p}_S - m_S \int^{\oplus} \frac{dm}{m} \left( \hat{p}_{A_L}^{(m)} \otimes \mathbb{1}_{A_R}^{(m*)} - \mathbb{1}_{A_L}^{(m)} \otimes \hat{p}_{A_R}^{(m*)} \right) \otimes \mathbb{1}_S, \quad (30a)$$

$$\hat{k}_{\overline{S|A}} = \hat{k}_A^R \otimes \mathbb{1}_S + \mathbb{1}_A \otimes \hat{k}_S - m_S \int^{\oplus} \frac{dm}{m} \left( \hat{k}_{A_L}^{(m)} \otimes \mathbb{1}_{A_R}^{(m*)} - \mathbb{1}_{A_L}^{(m)} \otimes \hat{k}_{A_R}^{(m*)} \right) \otimes \mathbb{1}_S, \quad (30b)$$

where $\hat{p}_A^R = \int^{\oplus} dm\, \mathbb{1}_{A_L}^{(m)} \otimes \hat{p}_{A_R}^{(m*)}$ and $\hat{p}_{A_R}^{(m*)}$ and $\hat{k}_{A_R}^{(m*)}$ are the generators of the complex-conjugate representation acting on $A_R$. The generators of the extra particle, $\hat{p}_{\overline{S|A}}$ and $\hat{k}_{\overline{S|A}}$, satisfy the commutation relations $\left[ \hat{p}_{\overline{S|A}}, \hat{k}_{\overline{S|A}} \right] = -i(\hat{M}_A \otimes \mathbb{1}_S + \mathbb{1}_A \otimes m_S \mathbb{1}_S)$, where $\hat{M}_A = \int^{\oplus} dm\, m\, \mathbb{1}_{A_L}^{(m)} \otimes \mathbb{1}_{A_R}^{(m*)}$. The reason for the minus sign in the commutation relation of the extra particle is a consequence of the commutation relations of the complex-conjugate representation $\hat{p}_{A_R}^{(m*)}$ and $\hat{k}_{A_R}^{(m*)}$, which satisfy $\left[ \hat{p}_{A_R}^{(m*)}, \hat{k}_{A_R}^{(m*)} \right] = -im\, \mathbb{1}_{A_R}^{m*}$.

Although $m$ can take, in principle, values over all $\mathbb{R}$, we can focus on the positive mass case by restricting the set of states on which our operators act. Let us now focus on a single mass sector of the regular representation, corresponding to mass $m > 0$. In Methods: Basis vectors for centrally extended Galilei group, we discuss normalisation issues that arise when restricting to a single mass sector. In what follows, it will be more instructive to deal with position operators instead of boost operators, so we write the boost operators in terms of position ones in Eq. (29). Thus, we focus on the momentum operator $\hat{p}_{S|A}^{(m)} = \mathbb{1}_A^{(m)} \otimes \hat{p}_S - (m_S/m)(\hat{p}_{A_L}^{(m)} \otimes \mathbb{1}_{A_R}^{(m*)} - \mathbb{1}_{A_L}^{(m)} \otimes \hat{p}_{A_R}^{(m*)}) \otimes \mathbb{1}_S$ and the position operator $\hat{x}_{S|A}^{(m)} = \mathbb{1}_A^{(m)} \otimes \hat{x}_S - \hat{x}_{A_L}^{(m)} \otimes \mathbb{1}_{A_R}^{(m*)} + \mathbb{1}_{A_L}^{(m)} \otimes \hat{x}_{A_R}^{(m*)} \otimes \mathbb{1}_S$, where $\mathbb{1}_A^{(m)} = \mathbb{1}_{A_L}^{(m)} \otimes \mathbb{1}_{A_R}^{(m*)}$.

We will now show that the system A can be seen as consisting of two particles, called $A_{m_1}$ and $A_{m_2}$ of respective masses $m_1$ and $m_2$, such that $m_1 + m_2 = m$, where $A_{m_1}$ serves as a reference for position and $A_{m_2}$ as a reference for velocity. We define $\hat{x}_{A_{m_1}} = (m\hat{x}_{A_L}^{(m)} \otimes \mathbb{1}_{A_R}^{(m*)} - m_2 \mathbb{1}_{A_L}^{(m)} \otimes \hat{x}_{A_R}^{(m*)})/2m_1$, and $\hat{x}_{A_{m_2}} = (m\hat{x}_{A_L}^{(m)} \otimes \mathbb{1}_{A_R}^{(m*)} + m_1 \mathbb{1}_{A_L}^{(m)} \otimes \hat{x}_{A_R}^{(m*)})/2m_2$. The momenta $\hat{p}_{A_{m_1}}$ and $\hat{p}_{A_{m_2}}$ are the conjugate variables to $\hat{x}_{A_{m_1}}$ and $\hat{x}_{A_{m_2}}$, respectively. In this way, $\hat{x}_{A_L}^{(m)} \otimes \mathbb{1}_{A_R}^{(m)}$ (the left-regular representation) can be seen as the position operator for the centre of mass of a system of our two particles, $A_{m_1}$ and $A_{m_2}$. That is, $\hat{x}_{A_L}^{(m)} \otimes \mathbb{1}_{A_R}^{(m)} = (m_1 \hat{x}_{A_{m_1}} + m_2 \hat{x}_{A_{m_2}})/m$. Similarly, the operator $\hat{p}_{A_L}^{(m)} \otimes \mathbb{1}_{A_R}^{(m*)}$ is the momentum of the centre of mass, $\hat{p}_{A_L}^{(m)} \otimes \mathbb{1}_{A_R}^{(m)} = \hat{p}_{A_{m_1}} + \hat{p}_{A_{m_2}}$. On the other hand, the operator $\mathbb{1}_{A_L}^{(m)} \otimes \hat{x}_{A_R}^{(m*)}$ (the right-regular representation) is proportional to the relative distance between $A_{m_1}$ and $A_{m_2}$, $\mathbb{1}_{A_L} \otimes \hat{x}_{A_R}^{(m*)} = (m_2/m)(\hat{x}_{A_{m_2}} - \hat{x}_{A_{m_1}})$, whereas $\mathbb{1}_{A_L}^{(m)} \otimes \hat{p}_{A_R}^{(m*)}$ corresponds to the relative momentum, $\mathbb{1}_{A_L}^{(m)} \otimes \hat{p}_{A_R}^{(m*)} = \hat{p}_{A_{m_1}} - (m_1/m_2)\hat{p}_{A_{m_2}}$.

Putting everything together, we arrive at

$$\hat{x}_{S|A} = \mathbb{1}_A \otimes \hat{x}_S - \hat{x}_{A_{m_1}} \otimes \mathbb{1}_S \quad (31a)$$

$$\hat{p}_{S|A} = \mathbb{1}_A \otimes \hat{p}_S - \frac{m_S}{m_2} \hat{p}_{A_{m_2}} \otimes \mathbb{1}_S, \quad (31b)$$

which expresses $\hat{x}_{S|A}$ and $\hat{p}_{S|A}$ as the position and momentum relative to two different particles, as we wanted to show.

We can also rewrite the algebra of the extra particle in terms of the two independent particles $A_{m_1}$ and $A_{m_2}$. For a single mass sector labeled by $m$, we plug the definition of $A_{m_1}$ and $A_{m_2}$ into Eq. (30), obtaining

$$\hat{x}_{\overline{S|A}} = \frac{1}{m_1 + m_2 + m_S}$$
$$\left( (m_2 + m_S)\hat{x}_{A_{m_1}} \otimes \mathbb{1}_S - m_2 \hat{x}_{A_{m_2}} \otimes \mathbb{1}_S - m_S \mathbb{1}_A \otimes \hat{x}_S \right) \quad (32a)$$

$$\hat{p}_{\overline{S|A}} = \hat{p}_{A_{m_1}} \otimes \mathbb{1}_S + \mathbb{1}_A \otimes \hat{p}_S - \frac{m_1 + m_S}{m_2} \hat{p}_{A_{m_2}} \otimes \mathbb{1}_S. \quad (32b)$$

Note that $\hat{p}_{\overline{S|A}}$ in Eq. (32b) is nothing elese than the relative momentum of particles $S$ and $A_{m_1}$ with respect to particle $A_{m_2}$.

If we have 2 QRFs, A and B, for the centrally extended Galilei group, the natural tensor product decompositions associated to A is related to the decomposition of B via Eqs. (24). In Methods: Relative generators for the centrally extended Galilei group, we compute explicitly the QRF transformation connecting the infinitesimal generators of the group "as seen" from QRF A to those "as seen" from QRF B.

In conclusion, the regular representation of the centrally extended Galilei Group can be seen as a system of variable mass, which under a properly normalised restriction to a fixed mass sector, consists of 2 particles, one of them serving as a QRF for position and the other as a QRF for velocity. These particles transform under the usual projective representation of the Galilei group. (In future work, it would be interested he formalism presented here can be extended to the case of projective representations). The case of a single mass sector with $m_1 = m_2 = m/2$ is depicted in Fig. 3.

### Comparison with other frameworks.

It is instructive to compare our framework in a given mass sector with other proposals for the relational description of multi-particle systems under Galilei and translation symmetries[7,9,37,38]. Assume that our reference frame A in the given mass sector is realised by particles 1 and 2 (we drop the label A for simplicity) serving as references for position and velocity, respetcively, and let the system S consist of $N$–2 particles, labeled by $i = 3, \cdots, N$. Denote the mass of particle $i$ by $m_i$ and the pair of its position and momentum operators in the standard partition by $(\hat{x}_i, \hat{p}_i)$, $i = 1, \cdots, N$. The Hilbert space of such an $N$-particle system defined relative to a hypothetical external observer decomposes as[9] $\mathcal{H} \cong \mathcal{H}_{CM} \otimes \mathcal{H}_{rel}$, where $\mathcal{H}_{CM}$ is the gauge subsystem corresponding to the centre of mass, defined by the position and momentum operators $x_{CM} = \sum_i m_i \hat{x}_i / M$, $P_{CM} = \sum_i \hat{p}_i$, where $M = \sum_i m_i$, and $\mathcal{H}_{rel}$ is the invariant subsystem containing relational degrees of freedom.

In our framework, the choice of particles 1 and 2 as a QRF gives rise to a decomposition of the invariant subsystem into a tensor product of the $N$–2 "system" particles defined relative to the QRF, plus the corresponding extra particle. The relative particles are given by the canonically conjugate pairs of relative position and momentum operators $(\hat{x}_{i|1}, \hat{p}_{i|2})$, where $\hat{x}_{i|1} = \hat{x}_i - \hat{x}_1$, and $\hat{p}_{i|2} = \hat{p}_i - \frac{m_i}{m_2}\hat{p}_2$, for $i = 3, \cdots, N$, and the extra particle by the canonically conjugate pair (30) restricted to the respective mass sector.

In comparison, ref. 9 considers only a single particle as a reference for either the position or velocity of the remaining particles. For example, if particle 1 is used as a reference for position, this is associated with a decomposition of the invariant subsystem into $N - 1$ relational particles, defined by the relative position operators $\hat{x}_{i|1} = \hat{x}_i - \hat{x}_1$, $i = 2, \cdots, N$ and canonically conjugate momenta $\hat{p}_{i|1} = \hat{p}_i - \frac{m_i}{M}\hat{p}_{CM}$. Note that, as seen from an external observer, the momenta in this case do not have an interpretation as the relative momenta of one particle relative to another, as the centre of mass is not a separate subsystem from such a perspective but rather it depends on the positions and masses of the whole collection of particles. In contrast, the relative momenta defined here depend only on the momenta of two particles: the momentum $\hat{p}_i$, $i = 3, \cdots, N$ and the momentum of the reference frame for velocity, particle 2.

Reference 37 has a completely internal treatment, where one "jumps" form the QRF of one internal observer to that of another one without invoking an external observer. It treats translations and Galilean boosts in 1 dimension as 2 separate cases, introducing a QRF transformation for translations and a different QRF transformation for boosts. Similar to refs. 7,9, ref. 37 uses a single-particle model of QRF. A single particle of finite mass $m$ can be either a perfect reference frame for the translation group, or a perfect reference frame for the group of Galilean boosts in one dimension, but not for both. In contrast, here we consider, in a fixed mass sector, a system of 2 particles serving as a QRF for both translations and boosts (which combined form the Galilei group in 1 dimension). Note that, in the limit $m \to \infty$, a single particle can serve as a perfect reference frame for both position and velocity. It would be

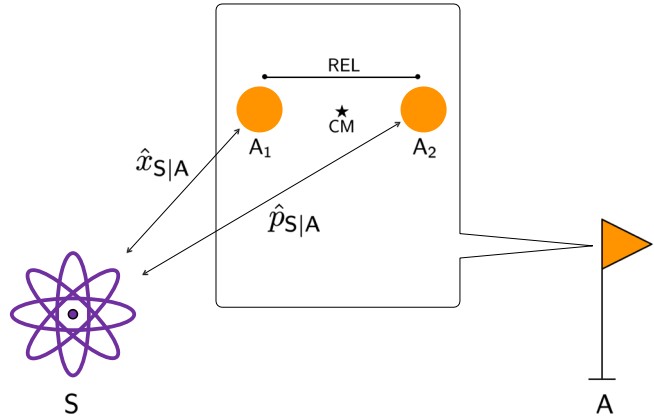

**Fig. 3 | Physical interpretation of the regular representation of the centrally extended Galilei Group.** For a given mass sector $m$, the regular representation can be seen as a system of 2 particles. Here we depict the case where each particle has a mass $m/2$. In this interpretation, the left regular representation corresponds to the degrees of freedom of the centre of mass, CM, of the 2-particle system. The right regular representation corresponds to the distance of any of the 2 particles to the centre of mass, or half their relative distance, REL. Imagine that Alice, using the reference frame A (orange flag), describes an operation on the system S (in purple). We can then ask how this operation "looks like" from the standard partition viewpoint. Roughly speaking, in this viewpoint, A uses one of the particles, $A_{m_1}$ (left orange circle), as a reference frame for position, and uses the other particle, $A_{m_2}$ (right orange circle), as a reference frame for velocity (see Eqs. (31)).

interesting to investigate the connection between this limit and the QRF model presented here.

Reference 38 obtains the QRF transformation for translations of ref. 37 by means of a gravity-inspired momentum constraint, which forces the centre-of-mass momentum of a "perspective neutral" state to vanish, $\hat{p}_{CM}|\Psi\rangle = 0$. Within the $\hat{p}_{CM} = 0$ subspace, the relational variables of refs. 37,38 are equivalent to that of[9]. However, the perspective-neutral state of ref. 38 does not have an immediate operational interpretation, as there is no external observer "out there" to measure such a state. Our framework is agnostic to whether such an external observer exists or not, and the constraint state $|\Psi\rangle$ can be interpreted as a state whose centre-of-mass momentum vanishes "as seen" by the external observer. This can be modelled in our framework by introducing an external reference frame for velocity, aligned with the velocity of the centre of mass. In this case, we would have a total of $N + 1$ particles, where particle $N + 1$ serves as a reference for velocity (and thereby momentum), while one of the other particles, say particle 1, serves as a reference for position. Note that Particle $N + 1$ in this example corresponds to what we have called particle $A_{m_2}$ in the previous subsection. The total relative momentum with respect to particle $N + 1$ thus coincides with the momentum of the extra particle given in Eq. (32b).

Ignoring the extra particle, and assuming that the total momentum of all particles from 1 to $N$ is zero relative to particle $N + 1$, we recover the description of refs. 37,38. In particular, the jumping transformations derived there can be understood as corresponding to changing which particle from 1 to $N$ serves as a reference for position, while keeping the reference for velocity fixed. As shown in Methods: Restriction to the zero-charge sector, our framework, restricted to the zero-charge sector for a general group reduces to the QRF transformation found in ref. 39. For the case of translations, this recovers formally the perspective-neutral computation of the QRF transformation developed in ref. 37.

In Methods: Compatibility with an external zero-charge state, we develop further the connection between our framework and the perspective neutral one by showing that, for compact groups, any state $\rho_{\overline{S|A},S|A}$ in Alice's viewpoint is consistent with the existence of a reference frame $\overline{B}$ and a pure state living in the zero-charge sector. Specifically, we show that if $\rho_{\overline{S|A},S|A}$ is an arbitrary (invariant) state of $\overline{S|A}$ and $S|A$ in Alice's perspective, then there is a pure state $|\phi\rangle_{ABS}$ in Eve's perspective such that

(i) $U_{ABS}(g)|\phi\rangle_{ABS} = |\phi\rangle_{ABS}$ for all $g$ (i.e., the state lives in the zero-charge subspace) and (ii) If we trace out B and "jump" to Alice's reference frame, we obtain $\rho_{\overline{S|A},S|A}$. Notice, however, that although such a "purification" in terms of the perspective-neutral framework is always possible, it is by no means necessary, as our framework contains the full invariant algebra of A and S. In fact, the extension to a zero-charge pure state is done at the expense of adding more (gauge and invariant) variables.

Finally, the work of Angelo et al.[7], proposed a relational description of particles within the invariant subsystem that uses a single particle, e.g., particle 1, as a reference for both position and velocity of the other particles, leading to a notion of relational particles with position and momentum operators $(\hat{x}_{i|1}, \hat{p}_{r_i})$, $i = 2, \cdots, N$, where $\hat{p}_{r_i} = \frac{m_1 m_r}{m_1 + m_r}(\frac{\hat{p}_i}{m_i} - \frac{\hat{p}_1}{m_1})$ (note that this notion of relative momentum is not equal to the relative velocity of the respective particle times its mass, but times the reduced mass of the particle and the reference, which is needed to ensure the canonical commutation relations for each particle). As emphasised in ref. 7, these particles are not separate systems, since their algebras do not commute with each other, and the canonical commutation relations are only recovered in the limit $m_1 \rightarrow \infty$.

The fact that $(\hat{x}_{i|1}, \hat{p}_{r_i})$, as defined by Angelo et al. are not a separate subsystems for different $i$ has drastic consequences, as ref. 7 illustrates by introducing the "paradox of the 3rd particle". In short, the paradox concerns the observation that, if one uses a single particle as a reference frame for both position and velocity in the context of Galilei symmetry, one arrives to the conclusion that the state of particle $S_2$ defined relative to particle $S_1$ depends on whether, relative to an external classical reference frame E (which can be modeled by a very heavy particle), there exists another particle $S_3$, separate from $S_1$ and $S_2$.

The resolution of the paradox proposed in ref. 7 is that two systems that are separate relative to E (in this case $S_2$ and $S_3$) may be overlapping when described relative to $S_1$, and therefore one cannot trace out $S_3$ from the state relative to $S_1$. Note, however, that this conclusion is obtained for a different model of QRF than the one we consider. In our framework, two separate systems are always separate relative to any QRF, and one can trace them out in any reference frame. Nevertheless, one should do this with care. As we have seen, when two observers using different QRFs refer to the same "system", they are referring to DOFs that belong to two different, albeit overlapping, subalgebras. Thus, in general their descriptions of the "system" would not contain the same information. Moreover, even if two observers Alice and Bob each describe the reference frame of the other in addition to the system S, the description of BS relative to Alice is given by an algebra BS|A that is not equal to the algebra AS|B describing AS relative to B. Thus, even in this case their descriptions would not contain the same information. The full invariant information, which is accessible by both observers, is only obtained when the extra particle is included in the description.

Recently, ref. 40 proposed a different analysis of the paradox of the third particle. They introduce a "relational partial trace" as a mathematical procedure for discarding subsystems, in an attempt to resolve the paradox in a gauge-independent way. That procedure, which has different operational grounds, leads to conclusions that are inequivalent to ours.

## Discussion
Symmetry transformations between QRFs can lead to a more general notion of symmetry in quantum mechanics, potentially sharpening our operational understanding of spacetime at the quantum level. For this reason, it is very important to understand what is at the root of the key differences between classical and quantum reference frame transformations. In this work, we have developed an approach to QRF transformations that focuses on the algebra of relative observables between a system and a reference frame. From this point of view, a QRF transformation is a change from a preferred tensor factorisation to another one. Moreover, given a set of QRFs, our approach fully characterises how different subsystem decompositions are connected to each other. This leads to a picture of the full invariant system of a quantum system as being composed by a network of subalgebras, with different parts of the network corresponding to different QRF viewpoints. A recent "perspective neutral" approach, presented in ref. 72, also develops QRF

transformations algebraically, for a large class of symmetry groups. While both approaches use similar mathematical techniques, our approach is applicable to the full invariant subsystem rather than to the zero-charge subspace. As discussed in the introduction, this implies a significant difference in the scope and physical interpretation of the two frameworks.

Our framework is naturally compatible with an incoherent-twirling approach to QRFs rather than a coherent twirling approach. This feature makes our approach a good candidate for studying QRF symmetries in a proper subsystem of the universe in the most general way. Approaches that restrict to a given charge subspace restrict the possible states in which the subsystem of the universe under study can be with respect to potential new systems out there. For this reason, a framework that focuses on a given charge subspace fails to capture the potential relation that the subsystem of interest might have with external degrees of freedom. Our approach, developed at the level of the full invariant subsystem of a given system, is compatible with extending the system we are interested in an arbitrary way. In this sense, our framework supports the view that, in some situations, incoherent twirling should be preferred to coherent twirling in the quantisation of systems with gauge symmetries[3,73].

There are several research avenues that our work opens. On the one hand, it would be very interesting to study QRF transformations with respect to relativistic groups, i.e., the Lorentz and Poincaré groups, and ask what operational notion of spacetime arises from such reference frames. These QRF transformations would allow us to study proper subsystems transforming under Lorentz and Poincaré symmetries, which is not thoroughly understood, and would be an important step towards ultimately incorporating general relativistic symmetries, which would be important in quantum gravity. We believe our approach is general and powerful enough to make such a study feasible. We note that there are recent examples in the quantum gravity literature that do not impose a total charge equal to zero in the Hamiltonian constraint; see, for example,[74].

On the other hand, we have focused on a very restrictive notion of quantum reference frame, namely, that corresponding to the regular representation of the group. The reason for doing this is to make explicit contact with our more familiar classical notion of reference frame viewpoints and transformations. Admittedly, the regular representation is a highly idealised object, and it would be very important to learn how to treat situations in which our reference frame is bounded in resources[3,7,8,73]. The first steps towards applying our framework to non-ideal frames appear in ref. 75. We believe the solution to this open problem can yield important insights beyond the approximation of superpositions of semiclassical causal structures and spacetimes, as for example the case considered in ref. 76.

## Methods
### Lagrangians for translation gauge symmetry
Here we give an example of two different Lagrangians that have the same gauge symmetry but impose different constraints. Our starting point is Eq. 57 of ref. 38. In the case $V = 0$, it reads

$$L = \frac{1}{2}\sum_{i=1}^{N}\dot{q}_i^2 - \frac{1}{2N}\left(\sum_{i=1}^{N}\dot{q}_i\right)^2. \tag{33}$$

This Lagrangian specifies the dynamics of an $N-$ particle system with coordinates $q_i$, $i = 1, \ldots N$. It is invariant under gauge transformations realised by time dependent translations, $q_i \rightarrow q_i + f(t)$, $\dot{q}_i \rightarrow \dot{q}_i + \dot{f}$. Computing the canonical momenta $p_i = \partial L/\partial \dot{q}_i$ and adding them up, we get a constraint on the total momentum, $P = \sum_i p_i = 0$.

At this point, it would seem that gauge invariance of the dynamics implies that the total momentum is constrained to be zero. However, there are more general possibilities. Consider the Lagrangian

$$L' = \frac{1}{2}\sum_{i=1}^{N}\dot{q}_i^2 - \frac{1}{2N}\left(\sum_{i=1}^{N}\dot{q}_i - \frac{p}{N}\right)^2. \tag{34}$$

It is easy to check that the new Lagrangian $L'$ is related to $L$ by a total derivative. Moreover, $L'$ has the same gauge symmetry as $L$. Under time-dependent translations, $L'$ changes by a total derivative.

Therefore, $L'$ is a completely valid Lagrangian under the lack of an external reference frame for spatial translations. However, $L'$ imposes a different constraint on the total momentum. We now have $P = \sum_i p_i = \sum_i \partial L'/\partial \dot{q} = p$.

We have constructed an example of two systems with the same gauge symmetry but different constraints on the total momentum, showing that a Lagrangian having a translation gauge symmetry ($q \to q + f(t)$) up to total derivatives does not necessarily impose a vanishing total momentum as a constraint. At the quantum level, this argument shows that the coherent gauge condition $\hat{P}|\Psi\rangle = 0$ comes from a specific choice of the Lagrangian ($L$ instead of $L'$ in our case), which imposes a constraint on a specific momentum sector.

As explained in the main text, the condition $\hat{P}|\Psi\rangle = 0$ can be interpreted in 2 ways: (1) $\hat{P}$ is not measurable and is set to vanish by convention. In this case it is clear that $P = k \neq 0$ is also a valid choice. This view is supported by our analisis of $L$ and $L'$ above. (2) $\hat{P}$ is measurable and found to vanish, as determined by some translation-invariant reference frame for momentum. If we admit the possibility that such translation-invariant reference frame for momentum could exist tacitly in our description, we should express the lack of a reference frame for translations in a way that makes no extra assumptions on the total charge. This is done by demanding $[\hat{P}, \rho] = 0$ instead of $\hat{P}|\Psi\rangle = 0$.

**Basis vectors for centrally extended Galilei group**

Here we compute the inner product between the basis vectors $|(\theta, a, v)\rangle$ of the regular representation of the centrally extended Galilei Group, and discuss the normalisation of states belonging to a single mass sector. For simplicity of notation, we omit the subscripts referring to the QRF perspective.

The Hilbert space of the regular representation of the centrally extended Galilei group is of the form

$$\mathcal{H} = \int^{\oplus} dm \, \mathcal{H}_\mathsf{L}^{(m)} \otimes \mathcal{H}_\mathsf{R}^{(m*)}, \tag{35}$$

where $\mathcal{H}_\mathsf{L}^{(m)}$ ($\mathcal{H}_\mathsf{R}^{(m*)}$) corresponds to the colour (flavour) degrees of freedom for mass $m$. The left-regular (right-regular) action of the group is trivial in $\mathcal{H}_\mathsf{R}^{(m*)}$ ($\mathcal{H}_\mathsf{L}^{(m)}$) for all $m$. For all $m$, the subspace $\mathcal{H}_\mathsf{L}^{(m)} \otimes \mathcal{H}_\mathsf{R}^{(m*)}$ is spanned by vectors of the form $|m; p, q\rangle$, satisfying $\langle m; p, q | m; p', q' \rangle = \delta(p - p')\delta(q - q')$. The labels $p$ and $(q)$ are eigenvalues of the momentum operator on $\mathcal{H}_\mathsf{L}^{(m)}$ ($\mathcal{H}_\mathsf{R}^{(m*)}$). The inner product of 2 states, $|\varphi\rangle = \int^{\oplus} dm \, |\varphi_m\rangle$ and $|\psi\rangle = \int^{\oplus} dm \, |\psi_m\rangle$ is defined by

$$\langle \varphi | \psi \rangle = \int dm \, \langle \varphi_m | \psi_m \rangle. \tag{36}$$

A normalised state $|\psi\rangle$ satisfies $\int dm \, \langle \psi_m | \psi_m \rangle = 1$.

By analogy with the compact group case of Eq. (5), vectors corresponding to a fixed group element $(\theta, a, v)$ are given by

$$|(\theta, a, v)\rangle = \int^{\oplus} dm \, dp \, \sqrt{m} \, (\tilde{U}^{(m)}(\theta, a, v)|m; p\rangle_\mathsf{L}) \otimes |m; p\rangle_\mathsf{R}, \tag{37}$$

where $\tilde{U}^{(m)}(\theta, a, v) = e^{i\theta} e^{-i((a+vt)\hat{p} - mv\hat{x})}$. We show that they are orthonormal in a generalised sense.

Using the Baker-Campbell-Hausdorff formula, we have

$$\tilde{U}^{(m)}(\theta, a, v) = e^{im(\theta + \frac{v}{2}(a+vt))} e^{-i(a+vt)\hat{p}} e^{imv\hat{x}}. \tag{38}$$

With the help of Eqs. (37) and (38), we can compute straightforwardly the inner product between 2 basis elements

$$
\begin{aligned}
\langle (\theta', a', v') | (\theta, a, v) \rangle &= \int dm \, dp \, m \, \langle m, p | \tilde{U}^{(m)} \left( \theta - \theta' - \frac{1}{2}(av' - a'v), a - a', v - v' \right) | m, p \rangle \\
&= \int dm \, dp \, m \, e^{im(s + \frac{\beta}{2}(\alpha + \beta t))} e^{-i(\alpha + \beta t)p} \delta(-m\beta),
\end{aligned}
\tag{39}
$$

where $s = \theta - \theta' - \frac{1}{2}(av' - a'v)$, $\alpha = a - a'$ and $\beta = v - v'$. Because for any normalised state there will always be an integral over $\beta$, we can use the identity $\delta(-m\beta) = \delta(\beta)/m$. Performing the integral over $p$, and going back to the original variables, the end result is

$$\langle (\theta', a', v') | (\theta, a, v) \rangle = \delta(\theta - \theta')\delta(a - a')\delta(v - v'), \tag{40}$$

as we wanted to show.

Given the full Hilbert space $\mathcal{H}$, how do we represent normalisable states on a single mass sector labeled by $m$? Because $m$ is a continuous parameter, we will only be able to do this in an approximate way. Consider the states $|\varphi_m^i\rangle = \int^{\oplus} dm' \, \sqrt{\Delta(m - m')} |\varphi_{m,m'}^i\rangle \in \mathcal{H}$, for $i$ running over a set of indices $\mathcal{I}$. Let $\{|\varphi_{m,m'}^i\rangle\}_{i \in \mathcal{I}}$ be an orthonormal basis on $\mathcal{H}_\mathsf{L}^{(m')} \otimes \mathcal{H}_\mathsf{R}^{(m'*)}$ for each $m'$, and $\Delta(m - m')$ be a sharply peaked function around $m$, such that we can approximate it by a Dirac delta, $\delta(m - m')$. Then, in this limit, we say that a normalisable state $|\psi_m\rangle \in \mathcal{H}$ belongs to the sector of mass $m$ if it is of the form $|\psi_m\rangle = \sum_i \psi_m^i |\varphi_m^i\rangle$, with $\sum_i |\psi_m^i|^2 = 1$. Then, formally, we can write the normalised basis states of a subspace of definite mass $m$ as

$$|\varphi_m^i\rangle = \int^{\oplus} dm' \, \sqrt{\delta(m - m')} |\varphi_{m,m'}^i\rangle, \tag{41}$$

and the projector onto the mass $m$ sector is given by

$$\Pi_m = \sum_i |\varphi_m^i\rangle \langle \varphi_m^i|. \tag{42}$$

Using the properties of the Dirac delta function, one can check that $\Pi_m^2 = \Pi_m$. For an explicit constructions of a set of functions converging to the square root of the delta function on three dimensions, see ref. 77.

**Group action in A's decomposition**

Here we prove Eq. (14). By direct calculation, we find

$$
\begin{aligned}
\mathcal{V}_{\mathsf{E} \to \mathsf{A}}(\hat{g}_\mathsf{A})[L_\mathsf{A}(g) \otimes U_\mathsf{S}(g)] &= \int dh' dh \, |h'\rangle \langle h'| L_\mathsf{C}(g)|h\rangle \langle h|_\mathsf{C} \otimes U_{\mathsf{S}|\mathsf{A}}^\dagger(h') U_{\mathsf{S}|\mathsf{A}}(g) U_{\mathsf{S}|\mathsf{A}}(h) \\
&= \int dh' dh \, |h'\rangle_\mathsf{C} \langle h'|gh\rangle \langle h|_\mathsf{C} \otimes U_{\mathsf{S}|\mathsf{A}}^\dagger(h') U_{\mathsf{S}|\mathsf{A}}(g) U_{\mathsf{S}|\mathsf{A}}(h) \\
&= \int dh \, L_\mathsf{C}(g)|h\rangle \langle h|_\mathsf{C} \otimes U_{\mathsf{S}|\mathsf{A}}^\dagger(gh) U_{\mathsf{S}|\mathsf{A}}(g) U_{\mathsf{S}|\mathsf{A}}(h) \\
&= L_\mathsf{C}(g) \otimes \mathbb{1}_{\mathsf{S}|\mathsf{A}},
\end{aligned}
\tag{43}
$$

Where we have used that $F_{\mathsf{E} \to \mathsf{A}}$ is its own inverse to change the labels in the first step. This proves Eq. (14).

**Independence of external frame**

Let Q be a generic quantum system (in the context of Subsection Modelling a quantum reference frame, Q = AS). Let E be an external reference frame with respect to which Q is described. We now show that, for any other possible external reference frame F describing Q, the invariant subalgebra $\mathcal{B}_{inv}(\mathcal{H}_\mathsf{Q})$ is the largest subalgebra common to both $\mathcal{B}(\mathcal{H}_{\mathsf{Q}|\mathsf{E}})$ and $\mathcal{B}(\mathcal{H}_{\mathsf{Q}|\mathsf{F}})$. This implies that $\mathcal{B}_{inv}(\mathcal{H}_\mathsf{Q})$ is independent of any potential external reference frame and at the same time compatible with any such reference frame,

in the sense that it is automatically a subalgebra of the larger invariant subalgebra arising from the addition of such a frame. It is easy to see that due to the assumed transversal action of the symmetry group, enlarging a given system by tensor-multiplying it with a new system always leads to a larger invariant subsystem that contains the old one as a subsystem.

Consider a Hilbert space containing all systems Q, E and F, $\mathcal{H}_{\mathsf{EFQ}} = \mathcal{H}_{\mathsf{E}} \otimes \mathcal{H}_{\mathsf{F}} \otimes \mathcal{H}_{\mathsf{Q}}$ (defined with respect to a yet more powerful observer). By definition, $\mathcal{B}(\mathcal{H}_{\mathsf{Q|E}})$ is formed by operators $T_{\mathsf{Q|E}}$ on $\mathcal{H}_{\mathsf{EFQ}}$ of the form

$$T_{\mathsf{Q|E}} = \int dg\, |g\rangle\,\langle g|_{\mathsf{E}} \otimes \mathbb{1}_{\mathsf{F}} \otimes U_{\mathsf{Q}}(g) T_{\mathsf{Q}}^{(\mathsf{E})} U_{\mathsf{Q}}^{\dagger}(g). \tag{44}$$

Analogously, $\mathcal{B}(\mathcal{H}_{\mathsf{Q|F}})$ is made of operators of the form

$$T_{\mathsf{Q|F}} = \int dg\, \mathbb{1}_{\mathsf{E}} \otimes |g\rangle\,\langle g|_{\mathsf{F}} \otimes U_{\mathsf{Q}}(g) T_{\mathsf{Q}}^{(\mathsf{F})} U_{\mathsf{Q}}^{\dagger}(g). \tag{45}$$

On the other hand, operators $T$ on $\mathcal{B}_{\mathrm{inv}}(\mathcal{H}_{\mathsf{Q}})$ have the form

$$T = \mathbb{1}_{\mathsf{E}} \otimes \mathbb{1}_{\mathsf{F}} \otimes T_{\mathsf{Q}}^{\mathrm{inv}}, \tag{46}$$

where $T_{\mathsf{Q}}^{\mathrm{inv}}$ is an invariant operator, $U_{\mathsf{Q}}(g) T_{\mathsf{Q}}^{\mathrm{inv}} U_{\mathsf{Q}}^{\dagger}(g) = T_{\mathsf{Q}}^{\mathrm{inv}}$ for all $g \in G$. From these definitions, it is clear that $\mathcal{B}_{\mathrm{inv}}(\mathcal{H}_{\mathsf{Q}})$ is a common subalgebra of $\mathcal{B}(\mathcal{H}_{\mathsf{Q|E}})$ and $\mathcal{B}(\mathcal{H}_{\mathsf{Q|F}})$, as Eq. (46) is a particular case of both Eqs. (44) and (45).

Now, let $T$ be a common element of both $\mathcal{B}(\mathcal{H}_{\mathsf{Q|E}})$ and $\mathcal{B}(\mathcal{H}_{\mathsf{Q|F}})$. We want to show that $T \in \mathcal{B}_{\mathrm{inv}}(\mathcal{H}_{\mathsf{Q}})$. For any Hilbert space carrying the left- and right-regular representations of $G$, define $|\Omega\rangle = \int dg|g\rangle$. If we set Eq. (44) equal to Eq. (45), multiply each side of the equality by $|\Omega\rangle\,\langle g|_{\mathsf{E}} \otimes |\Omega\rangle\,\langle h|_{\mathsf{F}} \otimes \mathbb{1}_{\mathsf{Q}}$ for arbitrary $g$ and $h$, and take the partial trace on E and F, we find that

$$U_{\mathsf{Q}}(g) T_{\mathsf{Q}}^{(\mathsf{E})} U_{\mathsf{Q}}^{\dagger}(g) = U_{\mathsf{Q}}(h) T_{\mathsf{Q}}^{(\mathsf{F})} U_{\mathsf{Q}}^{\dagger}(h), \tag{47}$$

for arbitrary $g$ and $h$. Setting $g = h$ gives $T_{\mathsf{Q}}^{(\mathsf{E})} = T_{\mathsf{Q}}^{(\mathsf{F})} = T_{\mathsf{Q}}$. Setting $g = e$ and $h$ arbitrary gives $T_{\mathsf{Q}} = U_{\mathsf{Q}}(h) T_{\mathsf{Q}} U_{\mathsf{Q}}^{\dagger}(h)$ for all $h$. This shows that $T \in \mathcal{B}_{\mathrm{inv}}(\mathcal{H}_{\mathsf{Q}})$.

### Explicit form of the $\mathcal{E}_{\mathsf{A}}$ transformation

Here we obtain an explicit form of the transformation $\mathcal{E}_{\mathsf{A}} = \mathcal{T}_{\mathsf{C}} \circ \mathcal{V}_{\mathsf{E} \to \mathsf{A}} = \mathcal{V}_{\mathsf{E} \to \mathsf{A}} \circ \mathcal{T}_{\mathsf{AS}}$ in the case of compact groups. Here, $\mathcal{T}_{\mathsf{AS}}$ is a superoperator projector onto the algebra of invariant (bounded) operators and $\mathcal{T}_{\mathsf{C}} = \mathcal{V}_{\mathsf{E} \to \mathsf{A}} \circ \mathcal{T}_{\mathsf{AS}} \circ \mathcal{V}_{\mathsf{E} \to \mathsf{A}}^{-1}$, where $\mathcal{V}_{\mathsf{E} \to \mathsf{A}}$ is defined above Eq. (13) in terms of the isomorphism $V_{\mathsf{E} \to \mathsf{A}}$, defined above Eq. (12). If $G$ is a compact group, $\mathcal{T}_{\mathsf{AS}}$ has a concrete representation in terms of the $G$-twirl, and we obtain

$$\begin{aligned}
\mathcal{E}_{\mathsf{A}}[T_{\mathsf{AS}}] &= \mathcal{V}_{\mathsf{E} \to \mathsf{A}} \circ \mathcal{T}_{\mathsf{AS}}[T_{\mathsf{AS}}] \\
&= U_{\mathsf{S|A}}^{\dagger}(\hat{g}_{\mathsf{C}}) \int dg L_{\mathsf{C}}(g) \otimes U_{\mathsf{S|A}}(g) T_{\mathsf{C,S|A}} L^{\dagger}(g)_{\mathsf{C}} \otimes U_{\mathsf{S|A}}^{\dagger}(g) \, U_{\mathsf{S|A}}(\hat{g}_{\mathsf{C}}) \\
&= \int dg L_{\mathsf{C}}(g) \otimes \mathbb{1}_{\mathsf{S|A}} U_{\mathsf{S|A}}^{\dagger}(\hat{g}_{\mathsf{C}}) T_{\mathsf{C,S|A}} U_{\mathsf{S|A}}(\hat{g}_{\mathsf{C}}) L_{\mathsf{C}}^{\dagger}(g) \otimes \mathbb{1}_{\mathsf{S|A}} \\
&= \mathcal{T}_{\mathsf{C}} \circ \mathcal{V}_{\mathsf{E} \to \mathsf{A}}[T_{\mathsf{AS}}].
\end{aligned} \tag{48}$$

To pass from the second to the third line, we have multiplied by the identity in the form $U_{\mathsf{S|A}}(\hat{g}_{\mathsf{C}}) U_{\mathsf{S|A}}^{\dagger}(\hat{g}_{\mathsf{C}})$ on the left of $T$ and in the form $U_{\mathsf{S|A}}^{\dagger}(\hat{g}_{\mathsf{C}}) U_{\mathsf{S|A}}(\hat{g}_{\mathsf{C}})$ on the right. Then we have used Eq. (14).

In the case of compact groups, for any operator $T_{\mathsf{AS}} = \int dg' dg |g'\rangle\,\langle g|_{\mathsf{A}} \otimes T_{\mathsf{S}}(g', g)$ in the standard partition, we can find its $G$-invariant version in the reference frame of Alice by applying the map $\mathcal{E}_{\mathsf{A}}$. The answer is

$$\mathcal{E}_{\mathsf{A}}[T] = \int dg' dg\, R_{\mathsf{C}}^{\dagger}(g') R_{\mathsf{C}}(g) \otimes U_{\mathsf{S|A}}^{\dagger}(g') T_{\mathsf{S|A}}(g', g) U_{\mathsf{S|A}}(g). \tag{49}$$

Proof:

$$\begin{aligned}
\mathcal{E}_{\mathsf{A}}[T_{\mathsf{AS}}] &= \mathcal{T}_{\mathsf{C}} \circ \mathcal{V}_{\mathsf{E} \to \mathsf{A}}[T_{\mathsf{AS}}] \\
&= \int dg' dg\, R_{\mathsf{C}}^{\dagger}(g') \int dh |h\rangle\,\langle h|_{\mathsf{C}} R_{\mathsf{C}}(g) \otimes U_{\mathsf{S|A}}^{\dagger}(g') T_{\mathsf{S|A}}(g', g) U_{\mathsf{S|A}}(g) \\
&= \int dg' dg\, R_{\mathsf{C}}^{\dagger}(g') R_{\mathsf{C}}(g) \otimes U_{\mathsf{S|A}}^{\dagger}(g') T_{\mathsf{S|A}}(g', g) U_{\mathsf{S|A}}(g).
\end{aligned} \tag{50}$$

### Exponential representation of quantum reference frame transformation

Here we derive a more intuitive expression for the quantum reference frame transformation of Eq. (19). We work at the operator rather than at the superoparator level. First, we establish a useful notation for our purposes. We write

$$V_{\mathsf{A} \to \mathsf{E}} = \int dg\, dh\, d\alpha\, |g\rangle_{\mathsf{A|E}}\langle g|_{\mathsf{C}} \otimes |h\rangle_{\mathsf{B|E}}\langle g^{-1}h|_{\mathsf{B|A}} \otimes |\alpha\rangle_{\mathsf{S|E}}\langle \alpha|_{\mathsf{S|A}} U_{\mathsf{S|A}}(g), \tag{51}$$

and

$$V_{\mathsf{B} \to \mathsf{E}}^{\dagger} = \int dg\, dh\, d\alpha\, |h^{-1}g\rangle_{\mathsf{A|B}}\langle g|_{\mathsf{A|E}} \otimes |h\rangle_{\mathsf{D}}\langle h|_{\mathsf{B|E}} \otimes U_{\mathsf{S|B}}^{\dagger}(h)|\alpha\rangle_{\mathsf{S|B}}\langle \alpha|_{\mathsf{S|E}}, \tag{52}$$

where the operators $V_{\mathsf{A} \to \mathsf{E}}$ and $V_{\mathsf{B} \to \mathsf{E}}^{\dagger}$ are defined below Eq. (19).

The quantum reference frame transformation is given by

$$S_{\mathsf{A} \to \mathsf{B}} = V_{\mathsf{B} \to \mathsf{E}}^{\dagger} V_{\mathsf{A} \to \mathsf{E}} \tag{53}$$

$$= \int dg\, dh\, d\alpha\, |g\rangle_{\mathsf{A|B}}\langle hg|_{\mathsf{C}} \otimes |h\rangle_{\mathsf{D}}\langle g^{-1}|_{\mathsf{B|A}} \otimes |\alpha\rangle_{\mathsf{S|B}}\langle \alpha|_{\mathsf{S|A}} U_{\mathsf{S|A}}(g). \tag{54}$$

Rearranging terms and using $\langle hg|_{\mathsf{C}} = \langle h|_{\mathsf{C}} R_{\mathsf{C}}(g)$, we find

$$\begin{aligned}
S_{\mathsf{A} \to \mathsf{B}} = \int dg\, dh\, d\alpha\, |g\rangle_{\mathsf{A|B}}\langle g^{-1}|_{\mathsf{B|A}} \otimes |h\rangle_{\mathsf{D}}\langle h|_{\mathsf{C}} \otimes |\alpha\rangle_{\mathsf{S|B}}\langle \alpha|_{\mathsf{S|A}} \\
\cdot \int df\, |f\rangle\,\langle f|_{\mathsf{B|A}} \otimes R_{\mathsf{C}}^{\dagger}(f) \otimes U_{\mathsf{S|A}}^{\dagger}(f).
\end{aligned} \tag{55}$$

Form the last expression it is manifest that $S_{\mathsf{A} \to \mathsf{B}}$ acts trivially on the gauge subsystem. (We say that a given operator acts trivially on a given subsystem defined by some subalgebra, if and only if the operator belongs to the comutant of that sublagebra). In the spirit of ref. 37, we now write the transformation in exponential form. We assume that $G$ is a Lie group such that for all $g \in G$ and for all representations $U$ of $G$ we have $U(g) = \exp(-i\lambda_g \cdot X)$. We can thus rewrite the second factor of Eq. (55) in exponential form, arriving at

$$S_{\mathsf{A} \to \mathsf{B}} = \mathcal{P}_{\mathsf{A} \to \mathsf{B}} e^{i \int dg \lambda_g |g\rangle\langle g|_{\mathsf{B|A}} \cdot \left( X_{\overline{\mathsf{BS|A}}} + X_{\mathsf{S|A}} \right)}, \tag{56}$$

where we have left tensor products with the identity operator implicit. Here

$$X_{\overline{\mathsf{BS|A}}} = \int^{\oplus} \mathbb{1}_{\mathsf{D_L}}^{(q)} \otimes X_{\mathsf{D_R}}^{(q)} \tag{57}$$

is the infinitesimal generator acting on the extra particle $\overline{\mathsf{BS|A}}$, in a notation consistent with Eq. (16). Note that $X_{\overline{\mathsf{BS|A}}}$ is a direct sum of the right-regular generators of the irreducible subspaces labeled by $q$, $X_{\mathsf{D_R}}^{(q)}$, with identity on the left-regular part, $\mathsf{D_L}$. Therefore, $X_{\overline{\mathsf{BS|A}}}$ commutes with any operator in the gauge subsystem, which corresponds to $\mathsf{D_L}$ in B's frame. On the other hand, $X_{\mathsf{S|A}}$ is the infinitesimal generator on the subsystem S|A. We have

defined the parity-swap operator $\mathcal{P}_{A \to B}$ as

$$\mathcal{P}_{A \to B} = \int dg\, dh\, d\alpha\, |g\rangle_{A|B} \langle g^{-1}|_{B|A} \otimes |h\rangle_D \langle h|_C \otimes |\alpha\rangle_{S|B} \langle \alpha|_{S|A}. \tag{58}$$

Alternatively, $\mathcal{P}_{A \to B}$ can be defined by its action on the subsystem $B|A$, acting trivially on all other subsystems:

$$\mathcal{P}_{A \to B} |g\rangle_{B|A} = |g^{-1}\rangle_{A|B}. \tag{59}$$

## Transformation of relative subsystems

Here we compute the transformation of observables from A to B. For operators in class 1, we have

$$
\begin{aligned}
\mathcal{S}_{A \to B}[\mathbb{1}_C \otimes \mathbb{1}_{B|A} \otimes T_{S|A}] &= \mathcal{V}_{E \to B} \circ \mathcal{V}_{E \to A}^\dagger[\mathbb{1}_C \otimes \mathbb{1}_{B|A} \otimes T_{S|A}] \\
&= \int dg'\, dg\, |g'g\rangle \langle g'g|_{A|B} \otimes |g'\rangle \langle g'|_D \otimes U_{S|B}(g'g) T_{S|B} U_{S|B}^\dagger(g'g) \\
&= \int dg\, |g\rangle \langle g|_{A|B} \otimes \mathbb{1}_D \otimes U_{S|B}(g) T_{S|B} U_{S|B}^\dagger(g).
\end{aligned}
\tag{60}
$$

For operators in class 2, we have

$$
\begin{aligned}
\mathcal{S}_{A \to B}[\mathbb{1}_C \otimes T_{B|A} \otimes \mathbb{1}_{S|A}] &= \mathcal{V}_{E \to B} \left[ \int dg\, |g\rangle \langle g|_A \otimes L_B(g) T_B L_B^\dagger(g) \otimes \mathbb{1}_S \right] \\
&= \int dh\, dg\, dh'\, |h^{-1}h'\rangle \langle g^{-1}h'|_{A|B} \otimes |h\rangle \langle (h')^{-1}h | T_D | (h')^{-1}g\rangle \langle g|_D \otimes U_{S|B}^\dagger(h'h) U_{S|B}(h'g) \\
&= \int dh\, dg\, |h^{-1}\rangle \langle h | T_{A|B} | g\rangle \langle g^{-1}|_{A|B} \otimes R_D(h^{-1}g) \otimes U_{S|B}(h^{-1}g),
\end{aligned}
\tag{61}
$$

where we have done the changes of variables $(h')^{-1}h \longrightarrow h$ and $(h')^{-1}g \longrightarrow g$ to pass from the second equality to the third one. Finally, for operators in class 3, we have

$$
\begin{aligned}
\mathcal{S}_{A \to B}[T_C^R \otimes \mathbb{1}_{B|A} \otimes \mathbb{1}_{S|A}] &= \mathcal{V}_{E \to B} \left[ \int dh'\, dh\, |h'\rangle \langle h' | T_A^R | h\rangle \langle h|_A \otimes L_B(h') L_B^\dagger(h) \otimes U_S(h') U_S^\dagger(h) \right] \\
&= \int dg\, dh'\, dh\, |g^{-1}\rangle \langle g^{-1}|_{A|B} \otimes R_D^\dagger(g)|h'\rangle \langle h' | T_D^R | h\rangle \langle h | R_D(g) \otimes \mathbb{1}_{S|B} \\
&= \int dg\, |g\rangle \langle g|_{A|B} \otimes R_D(g) T_D^R R_D^\dagger(g) \otimes \mathbb{1}_{S|B}.
\end{aligned}
\tag{62}
$$

This proves Eqs. (24).

## Restriction to the zero-charge sector

Here we consider the special case of the zero-charge sector of the invariant subspace, and show how to formally obtain the transformation rule of [39] from our framework. The derivation follows the perspective neutral framework [38] applied to a general group $G$.

Consider 2 reference frames A and B and a quantum system S. As in the main text, the total Hilbert space decomposes into a sum of charge sectors. Suppose we have a quantum state $|\Psi\rangle$ in the zero-charge sector of the total Hilbert space. In the standard partition, such a state satisfies $L_A(g) \otimes L_B(g) \otimes U_S(g)|\Psi\rangle = |\Psi\rangle$ for all $g \in G$. Note that this condition is strictly stronger than requiring the invariance of the density matrix $\rho$ under the action of $G$: $L_A(g) \otimes L_B(g) \otimes U_S(g)\rho L_A^\dagger(g) \otimes L_B^\dagger(g) \otimes U_S^\dagger(g) = \rho$. The state $|\Psi\rangle$ can be obtained by "coherent group averaging" over an arbitrary state $|\varphi\rangle = \int dg_A\, dg_B\, |g_A\rangle_A \otimes |g_B\rangle_B \otimes |\varphi(g_A, g_B)\rangle_S$. Then we have

$$|\Psi\rangle = \int dg\, L_A(g) \otimes L_B(g) \otimes U_S(g)|\varphi\rangle. \tag{63}$$

As in the main text, the state in the partition natural to A is found by applying $U_{BS}^\dagger(\hat{g}_{sfA})$ on $|\Psi\rangle$. The result is that the state of the reference frame A factors out for any initial state $|\varphi\rangle$. That is

$$U_{BS}^\dagger(\hat{g}_A)|\Psi\rangle = |\Omega\rangle_C \otimes \int dg\, L_{B|A}^\dagger(g) \otimes U_{S|A}^\dagger(g)|\varphi(g)\rangle_{B|A, S|A}, \tag{64}$$

where $|\Omega\rangle = \int dg\, |g\rangle$, as in Methods: Independence of external frame, and $|\varphi(g)\rangle_{B|A, S|A} = \int dg'\, |g'\rangle_{B|A} \otimes |\varphi(g, g')\rangle_{S|A}$. We interpret $\int dg\, L_{B|A}^\dagger(g) \otimes U_{S|A}^\dagger(g)|\varphi(g)\rangle_{B|A, S|A}$ as the state of B and S "as seen" form A.

By construction, applying the operator $S_{A \to B} = U_{AS}^\dagger(\hat{g}_B) U_{BS}(\hat{g}_A)$ to $U_{BS}^\dagger(\hat{g}_A)|\Psi\rangle$ gives

$$
\begin{aligned}
U_{AS}^\dagger(\hat{g}_B)|\Psi\rangle = \int dg\, dg'\, L_{A|B}(g)^\dagger \otimes \mathbb{1}_D \otimes U_{B|S}^\dagger(g)|g'\rangle_{A|B} \otimes \\
|\Omega\rangle_D \otimes |\varphi(g', g)\rangle_{S|B},
\end{aligned}
\tag{65}
$$

Which is the analogue of Eq. (64) with B playing the role of A. Now define

$$\hat{D} = \text{SWAP}_{AB} \circ \mathbb{1}_C \otimes \int dh\, |h^{-1}\rangle \langle h|_{B|A} \otimes U_{S|A}^\dagger(h), \tag{66}$$

where $\text{SWAP}_{AB}$ is the operator that swaps A and B's Hilbert spaces. A straightforward calculation gives

$$\hat{D} U_{BS|A}^\dagger(\hat{g}_A)|\Psi\rangle = U_{AS|B}^\dagger(\hat{g}_B)|\Psi\rangle, \tag{67}$$

showing that $S_{A \to B}$ and $\hat{D}$ coincide in the zero-charge subspace. The operator $\hat{D}$ is the one found in ref. [39] up to an arbitrary exchange of the roles between the left- and right-regular representations. In ref. [39], the state associated to the reference frame whose perspective we "jump" into is the neutral element of the group $e$. This can be fixed in the present perspective, up to normalisation, by conditioning the state of the reference frame to be $|e\rangle$ [38]. In conclusion, we have shown that our results formally reduce to those of [39] in the zero-charge subspace.

## Extra particle and unitarity

To appreciate the importance of the extra particle for obtaining a unitary (passive) transformation in Bob's description when Bob's reference frame is subject to a unitary (active) transformation relative to Alice, consider a simple scenario. Let A be in a classical state and let B be in the state $|e\rangle \langle e|_{B|A}$ in the perspective of A. This means that A is also in the state $|e\rangle \langle e|_{A|B}$ in the perspective of B (i.e., the two reference frames are aligned).

Let S be in some pure state $|\psi\rangle \langle \psi|$, which would be the same in both perspectives, i.e., we have $|\psi\rangle \langle \psi|_{S|A}$ and $|\psi\rangle \langle \psi|_{S|B}$. If now a unitary is applied in $B|A$, taking the state of B relative to A to a nontrivial superposition of group states, $|\phi\rangle \langle \phi|_{B|A}$, where $|\phi\rangle_{B|A} = \int dg\, \phi(g)|g\rangle_{B|A}$, such that this state is not invariant under the action of the group, it is easy to see that Bob would describe the system and reference frame of Alice by the

mixed state

$$\rho_{\text{AS|B}} = \int dg \, |\phi(g)|^2 |g^{-1}\rangle\langle g^{-1}|_{\text{A|B}} \otimes U^\dagger(g)_{\text{S|B}} |\psi\rangle\langle\psi|_{\text{S|B}} U(g)_{\text{S|B}}, \quad (68)$$

which cannot be unitarily related to the initial pure state $|e\rangle\langle e|_{\text{A|B}} \otimes |\psi\rangle\langle\psi|_{\text{S|B}}$. In other words, some information has been lost.

A naive attempt to recover this information by searching for it in the rest of the universe outside of A and B can be immediately seen to fail since any system S outside of A and B will be in an analogous classical correlation with A from the perspective of Bob. The resolution to this apparent paradox is that the state of AS|B is purified on the extra particle $\overline{\text{AS|B}}$, which is inside the invariant subsystems of ABS. This should come as no surprise since the active unitary transformation we considered was confined within this invariant subsystem.

One may nevertheless ask how come any other system in the rest of the universe gets correlated with A in the perspective of Bob if the transformation is so confined. The answer is that even thought other systems in the perspective of Bob correspond to separate subsystems, such A|B, S|B, etc., each of these subsystems overlaps with the subsystem B|A on which the unitary acts (in the sense that the corresponding algebras do not commute), hence they would generally all be affected.

### Relative generators for the centrally extended Galilei group

Here we derive Eqs. (29) of the main text. We work in the standard partition. By definition (Eq. (8)),

$$\hat{p}_{\text{S|A}} = \int d\theta da d\nu \, |(\theta, a, \nu)\rangle\langle(\theta, a, \nu)|_{\text{A}} \otimes \tilde{U}_{\text{S}}^{(m_{\text{S}})}(\theta, a, \nu) \hat{p}_{\text{S}} \tilde{U}_{\text{S}}^{(m_{\text{S}})\dagger}(\theta, a, \nu), \quad (69a)$$

$$\hat{k}_{\text{S|A}} = \int d\theta da d\nu \, |(\theta, a, \nu)\rangle\langle(\theta, a, \nu)|_{\text{A}} \otimes \tilde{U}_{\text{S}}^{(m_{\text{S}})}(\theta, a, \nu) \hat{k}_{\text{S}} \tilde{U}_{\text{S}}^{(m_{\text{S}})\dagger}(\theta, a, \nu). \quad (69b)$$

A straightforward calculation of the second tensor factor in Eqs. (69) gives

$$\hat{p}_{\text{S|A}} = \mathbb{1}_{\text{A}} \otimes \hat{p}_{\text{S}} - m_{\text{S}} \hat{v}_{\text{A}}^{\text{reg.}} \otimes \mathbb{1}_{\text{S}}, \quad (70a)$$

$$\hat{k}_{\text{S|A}} = \mathbb{1}_{\text{A}} \otimes \hat{k}_{\text{S}} + m_{\text{S}} \hat{a}_{\text{A}}^{\text{reg.}} \otimes \mathbb{1}_{\text{S}}, \quad (70b)$$

where

$$\hat{a}_{\text{A}}^{\text{reg.}} = \int d\theta da d\nu \, a \, |(\theta, a, \nu)\rangle\langle(\theta, a, \nu)|_{\text{A}}, \quad (71)$$

$$\hat{v}_{\text{A}}^{\text{reg.}} = \int d\theta da d\nu \, \nu \, |(\theta, a, \nu)\rangle\langle(\theta, a, \nu)|_{\text{A}}. \quad (72)$$

(The superscript reg. stands for "regular", as in the regular representation). Therefore, it all amounts to calculating $\hat{a}_{\text{A}}^{\text{reg.}}$ and $\hat{v}_{\text{A}}^{\text{reg.}}$.

Let us start by expressing the projector $|(\theta, a, \nu)\rangle\langle(\theta, a, \nu)|_{\text{A}}$ in the basis of irreducible representations of $\tilde{G}$. As in the main text, we denote the left-regular subsystem of A by $\text{A}_{\text{L}}$ and the right-regular subsystem by $\text{A}_{\text{R}}$. For the case of the (noncompact) extended Galilei group, the analogue to Eq. (5) is

$$|(\theta, a, \nu)\rangle_{\text{A}} = \int^\oplus dm dp \, \sqrt{m} \, (\tilde{U}^{(m)}(\theta, a, \nu)|m; p\rangle_{\text{A}_{\text{L}}}) \otimes |m; p\rangle_{\text{A}_{\text{R}}}, \quad (73)$$

where $\hat{p}_{\text{A}_{\text{L}}}|m; p\rangle_{\text{A}_{\text{L}}} = p|m; p\rangle_{\text{A}_{\text{L}}}$. Note the presence of the factor $\sqrt{m}$, analogue to $\dim(q)/|\mathcal{G}|$, which ensures that the normalisation condition is met (see Methods: Basis vectors for centrally extended Galilei group). With this

identity at hand, together with $\hat{k} = t\hat{p} - m\hat{x}$, we can write

$$|(\theta, a, \nu)\rangle\langle(\theta, a, \nu)|_{\text{A}} = \int^\oplus dm dm' \Big( m \, e^{i\theta(m-m')} e^{-i(a+\nu t)(p+\frac{1}{2}m\nu - p' - \frac{1}{2}m'\nu)}$$
$$|m; p+m\nu\rangle\langle m'; p'+m'\nu|_{\text{A}_{\text{L}}} \otimes |m; p\rangle\langle m'; p'|_{\text{A}_{\text{R}}} \Big). \quad (74)$$

The only dependence on $\theta$ in both $\hat{a}_{\text{reg.}}$ and $\hat{v}_{\text{reg.}}$ comes from the first exponential in Eq. (74). This means we can perform the integral over $\theta$ straight away, leading to a superselection on the mass. (We neglect factors of $\pi$ when using the Fourier transform of the Dirac delta function).

Doing the change of variable $a + \nu t \longrightarrow a$, it follows by direct calculation that $\hat{a}_{\text{A}}^{\text{reg.}} = \hat{a}'^{\text{reg.}}_{\text{A}} - t \, \hat{v}_{\text{A}}^{\text{reg.}}$, where

$$\hat{a}'^{\text{reg.}}_{\text{A}} = \int^\oplus da d\nu dm dp dp' \, ma \, e^{-ia(p-p')}|p+m\nu\rangle\langle p'+m\nu|_{\text{A}_{\text{L}}} \otimes |p\rangle\langle p'|_{\text{A}_{\text{R}}} \quad (75)$$

At this point, the integral $\hat{v}_{\text{A}}^{\text{reg.}}$ follows immediately, resulting in

$$\hat{v}_{\text{A}}^{\text{reg.}} = \int^\oplus \frac{dm}{m} (\hat{p}_{\text{A}_{\text{L}}}^{(m)} \otimes \mathbb{1}_{\text{A}_{\text{R}}}^{(m*)} - \mathbb{1}_{\text{A}_{\text{L}}}^{(m)} \otimes \hat{p}_{\text{A}_{\text{R}}}^{(m*)}). \quad (76)$$

For each irrep, labeled by $m$, both the left-regular and the right-regular operators $\hat{p}_{\text{A}_{\text{L}}}^{(m)} \otimes \mathbb{1}_{\text{A}_{\text{R}}}^{(m*)}$ and $\mathbb{1}_{\text{A}_{\text{L}}}^{(m)} \otimes \hat{p}_{\text{A}_{\text{R}}}^{(m*)}$ are present in Eq. (76). Note that this integral has a block diagonal structure due to superselection on the mass.

Finally, using $|m; p+m\nu\rangle_{\text{A}_{\text{L}}} = e^{-i\frac{p}{m}\hat{x}}|m; m\nu\rangle_{\text{A}_{\text{L}}}$, we can compute $\hat{a}'^{\text{reg.}}_{\text{A}}$. The result is

$$\hat{a}'^{\text{reg.}}_{\text{A}} = -\int^\oplus \frac{dm}{m} (\hat{k}_{\text{A}_{\text{L}}}^{(m)} \otimes \mathbb{1}_{\text{A}_{\text{R}}}^{(m*)} - \mathbb{1}_{\text{A}_{\text{L}}}^{(m)} \otimes \hat{k}_{\text{A}_{\text{R}}}^{(m*)}). \quad (77)$$

Putting all the pieces together (at $t = 0$), we arrive at

$$\hat{p}_{\text{S|A}} = \mathbb{1}_{\text{A}} \otimes \hat{p}_{\text{S}} - m_{\text{S}} \int^\oplus \frac{dm}{m} (\hat{p}_{\text{A}_{\text{L}}}^{(m)} \otimes \mathbb{1}_{\text{A}_{\text{R}}}^{(m*)} - \mathbb{1}_{\text{A}_{\text{L}}}^{(m)} \otimes \hat{p}_{\text{A}_{\text{R}}}^{(m*)}) \otimes \mathbb{1}_{\text{S}}, \quad (78a)$$

$$\hat{k}_{\text{S|A}} = \mathbb{1}_{\text{A}} \otimes \hat{k}_{\text{S}} - m_{\text{S}} \int^\oplus \frac{dm}{m} (\hat{k}_{\text{A}_{\text{L}}}^{(m)} \otimes \mathbb{1}_{\text{A}_{\text{R}}}^{(m*)} - \mathbb{1}_{\text{A}_{\text{L}}}^{(m)} \otimes \hat{k}_{\text{A}_{\text{R}}}^{(m*)}) \otimes \mathbb{1}_{\text{S}}, \quad (78b)$$

which are Eqs. (29).

Let us now compute the generators of the extra particle, $\hat{p}_{\overline{\text{S|A}}}$ and $\hat{k}_{\overline{\text{S|A}}}$. To do this, we use a trick that is valid for QRFs associated to arbitrary Lie groups $G$. Let $R_{\text{A}}(\delta) = \int^\oplus dq \, \mathbb{1}_{\text{A}_{\text{L}}}^{(q)} \otimes D_{\text{A}_{\text{R}}}^{(q*)}(\delta)$ be the right-regular representation of a group element $\delta$ (for simplicity of notation, we write $R_{\text{A}}(\delta)$ instead of $R_{\text{A|E}}(\delta)$). Assume $\delta$ is such that, for every value of the charge $q$, we can write $D_{\text{A}_{\text{R}}}^{(q*)}(\delta) = e^{i\epsilon_\delta \cdot X_{\text{A}_{\text{R}}}^{(q*)}}$ for a parametrisation of $\delta$ given by $\epsilon_\delta$ and an infinitesimal generator $X_{\text{A}_{\text{R}}}^{(q*)}$ By the orthogonality of the subspaces corresponding to different $q$'s, we can write

$$R_{\text{A}}(\delta) = \int^\oplus dq \, \mathbb{1}_{\text{A}_{\text{L}}}^{(q)} \otimes e^{i\epsilon_\delta \cdot X_{\text{A}_{\text{R}}}^{(q*)}} = e^{i\epsilon_\delta \cdot X_{\text{A}}^{\text{R*}}}, \quad (79)$$

where $X_{\text{A}}^{\text{R*}} = \int^\oplus dq \, \mathbb{1}_{\text{A}_{\text{L}}}^{(q)} \otimes X_{\text{A}_{\text{R}}}^{(q*)}$. Let us expand $R_{\text{A}}(\delta)$ to first order in a Taylor series around $\epsilon_\delta$, so that $R_{\text{A}}(\delta) = \mathbb{1}_{\text{A}} + i\epsilon_\delta \cdot X_{\text{A}}^{\text{R*}} + \cdots$. Now we can use this representation of $R_{\text{A}}(\delta)$ to compute $X_{\overline{\text{S|A}}}$ from Eq. (18). There are 2 ways in which we can compute the right-hand side of Eq. (18) for the case of $R_{\text{A}}(\delta)$. We can expand to first order in $\epsilon_\delta$ and then compute the integral, or we can first compute the integral and then expand to first order in $\epsilon_\delta$.

Equating the order $\epsilon_\delta$ of both Taylor series gives

$$X_{\overline{S|A}} = X_{A}^{R*} \otimes \mathbb{1}_S + \mathbb{1}_A \otimes X_S. \tag{80}$$

Note that both $X_{A}^{R*} \otimes \mathbb{1}_S$ and $\mathbb{1}_A \otimes X_S$ have an overall positive sign in Eq. (80). This is because the right-regular representation is defined in terms of the complex-conjugate representations $D_{A_R}^{(q*)} = e^{i\epsilon_\delta \cdot X_{A_R}^{(q*)}}$, whereas $U_S = e^{-i\epsilon_\delta \cdot X_S}$.

Applying Eq. (80) to the case of the centrally extended Galilei group gives immediately

$$\hat{p}_{\overline{S|A}} = \hat{p}_{A}^{R} \otimes \mathbb{1}_S + \mathbb{1}_A \otimes \hat{p}_S - m_S \int^\oplus \frac{dm}{m} (\hat{p}_{A_L}^{(m)} \otimes \mathbb{1}_{A_R}^{(m*)} - \mathbb{1}_{A_L}^{(m)} \otimes \hat{p}_{A_R}^{(m*)}) \otimes \mathbb{1}_S, \tag{81a}$$

$$\hat{k}_{\overline{S|A}} = \hat{k}_{A}^{R} \otimes \mathbb{1}_S + \mathbb{1}_A \otimes \hat{k}_S - m_S \int^\oplus \frac{dm}{m} (\hat{k}_{A_L}^{(m)} \otimes \mathbb{1}_{A_R}^{(m*)} - \mathbb{1}_{A_L}^{(m)} \otimes \hat{k}_{A_R}^{(m*)}) \otimes \mathbb{1}_S, \tag{81b}$$

which are Eqs. (30).

For completeness, let us write down explicitly Eqs. (24) for the infinitesimal generators of the centrally Extended Galilei group. The case of Eq. (24a) is straightforward from the computation of the algebra S|A, as one only needs to add an extra identity operator in the Hilbert space of B. We have already computed this algebra for the centrally extended Galilei group (see Eqs. (29) and Methods: Relative generators for the centrally extended Galilei group). The result is

$$\mathcal{S}_{A\to B}[\mathbb{1}_C \otimes \mathbb{1}_{B|A} \otimes \hat{p}_{S|A}]$$
$$= \mathbb{1}_{A|B} \otimes \mathbb{1}_D \otimes \hat{p}_{S|B} - m_S \int^\oplus \frac{dm}{m} (\hat{p}_{A|B_L}^{(m)} \otimes \mathbb{1}_{A|B_R}^{(m*)} - \mathbb{1}_{A|B_L}^{(m)} \otimes \hat{p}_{A|B_R}^{(m*)}) \otimes \mathbb{1}_D \otimes \mathbb{1}_{S|B}, \tag{82a}$$

$$\mathcal{S}_{A\to B}[\mathbb{1}_C \otimes \mathbb{1}_{B|A} \otimes \hat{k}_{S|A}]$$
$$= \mathbb{1}_{A|B} \otimes \mathbb{1}_D \otimes \hat{k}_{S|B} - m_S \int^\oplus \frac{dm}{m} (\hat{k}_{A|B_L}^{(m)} \otimes \mathbb{1}_{A|B_R}^{(m*)} - \mathbb{1}_{A|B_L}^{(m)} \otimes \hat{k}_{A|B_R}^{(m*)}) \otimes \mathbb{1}_D \otimes \mathbb{1}_{S|B}. \tag{82b}$$

Expressing the right-regular action in exponential form, as we did in the derivation that led to Eq. (80), the case of Eq. (24c) follows in essentially the same way as the case of Eq. (24a), giving

$$\mathcal{S}_{A\to B}[\hat{p}_{C}^{R} \otimes \mathbb{1}_{B|A} \otimes \mathbb{1}_{S|A}]$$
$$= \mathbb{1}_{A|B} \otimes \hat{p}_{D}^{R} \otimes \mathbb{1}_{S|B} - \int^\oplus \frac{dm}{m} (\hat{p}_{A|B_L}^{(m)} \otimes \mathbb{1}_{A|B_R}^{(m*)} - \mathbb{1}_{A|B_L}^{(m)} \otimes \hat{p}_{A|B_R}^{(m*)}) \otimes \hat{M}_D \otimes \mathbb{1}_{S|B}, \tag{83a}$$

$$\mathcal{S}_{A\to B}[\hat{k}_{C}^{R} \otimes \mathbb{1}_{B|A} \otimes \mathbb{1}_{S|A}]$$
$$= \mathbb{1}_{A|B} \otimes \hat{k}_{D}^{R} \otimes \mathbb{1}_{S|B} - \int^\oplus \frac{dm}{m} (\hat{k}_{A|B_L}^{(m)} \otimes \mathbb{1}_{A|B_R}^{(m*)} - \mathbb{1}_{A|B_L}^{(m)} \otimes \hat{k}_{A|B_R}^{(m*)}) \otimes \hat{M}_D \otimes \mathbb{1}_{S|B}. \tag{83b}$$

Finally, we can compute the case of Eq. (24b) by means of a similar trick to that leading to Eq. (80). That is, we can compute Eq. (24b) in two equivalent ways and equate the results. In the first way, we solve the integrals in Eq. (24b) for an infinitesimal transformation and then expand the result to first order in the parameter multiplying the generator. In the second way, we expand first and write down the integrals afterwards. Following this

technique for the generators of the centrally extended Galilei group yields

$$\mathcal{S}_{A\to B}[\mathbb{1}_C^R \otimes \hat{p}_{B|A}^{R*} \otimes \mathbb{1}_{S|A}]$$
$$= -\hat{p}_{A|B}^{L} \otimes \mathbb{1}_D \otimes \mathbb{1}_{S|B} + \mathbb{1}_{A|B} \otimes \hat{p}_{D}^{R*} \otimes \mathbb{1}_{S|B} - \mathbb{1}_{A|B} \otimes \mathbb{1}_D \otimes \hat{p}_{S|B}, \tag{84a}$$

$$\mathcal{S}_{A\to B}[\mathbb{1}_C^R \otimes \hat{k}_{B|A}^{R*} \otimes \mathbb{1}_{S|A}]$$
$$= -\hat{k}_{A|B}^{L} \otimes \mathbb{1}_D \otimes \mathbb{1}_{S|B} + \mathbb{1}_{A|B} \otimes \hat{k}_{D}^{R*} \otimes \mathbb{1}_{S|B} - \mathbb{1}_{A|B} \otimes \mathbb{1}_D \otimes \hat{k}_{S|B}, \tag{84b}$$

$$\mathcal{S}_{A\to B}[\mathbb{1}_C^R \otimes \hat{M}_{B|A} \otimes \mathbb{1}_{S|A}]$$
$$= -\hat{M}_{A|B} \otimes \mathbb{1}_D \otimes \mathbb{1}_{S|B} + \mathbb{1}_{A|B} \otimes \hat{M}_D \otimes \mathbb{1}_{S|B} - \mathbb{1}_{A|B} \otimes \mathbb{1}_D \otimes m_S \mathbb{1}_{S|B}, \tag{84c}$$

and

$$\mathcal{S}_{A\to B}[\mathbb{1}_C^R \otimes \hat{p}_{B|A}^{L} \otimes \mathbb{1}_{S|A}] = -\hat{p}_{A|B}^{R*} \otimes \mathbb{1}_D \otimes \mathbb{1}_{S|B} + \mathbb{1}_{A|B} \otimes \hat{p}_{D}^{R*} \otimes \mathbb{1}_{S|B} - \mathbb{1}_{A|B} \mathbb{1}_D \otimes \hat{p}_{S|B}$$
$$- \int^\oplus \frac{dm}{m} (\hat{p}_{A|B_L}^{(m)} \otimes \mathbb{1}_{A|B_R}^{(m*)} - \mathbb{1}_{A|B_L}^{(m)} \otimes \hat{p}_{A|B_R}^{(m*)}) \otimes \hat{M}_D \otimes \mathbb{1}_{S|B}$$
$$+ \int^\oplus \frac{dm}{m} (\hat{p}_{A|B_L}^{(m)} \otimes \mathbb{1}_{A|B_R}^{(m*)} - \mathbb{1}_{A|B_L}^{(m)} \otimes \hat{p}_{A|B_R}^{(m*)}) \otimes \mathbb{1}_D \otimes m_S \mathbb{1}_{S|B}, \tag{85a}$$

$$\mathcal{S}_{A\to B}[\mathbb{1}_C^R \otimes \hat{k}_{B|A}^{L} \otimes \mathbb{1}_{S|A}] = -\hat{k}_{A|B}^{R*} \otimes \mathbb{1}_D \otimes \mathbb{1}_{S|B} + \mathbb{1}_{A|B} \otimes \hat{k}_{D}^{R*} \otimes \mathbb{1}_{S|B} - \mathbb{1}_{A|B} \mathbb{1}_D \otimes \hat{k}_{S|B}$$
$$- \int^\oplus \frac{dm}{m} (\hat{k}_{A|B_L}^{(m)} \otimes \mathbb{1}_{A|B_R}^{(m*)} - \mathbb{1}_{A|B_L}^{(m)} \otimes \hat{k}_{A|B_R}^{(m*)}) \otimes \hat{M}_D \otimes \mathbb{1}_{S|B}$$
$$+ \int^\oplus \frac{dm}{m} (\hat{k}_{A|B_L}^{(m)} \otimes \mathbb{1}_{A|B_R}^{(m*)} - \mathbb{1}_{A|B_L}^{(m)} \otimes \hat{k}_{A|B_R}^{(m*)}) \otimes \mathbb{1}_D \otimes m_S \mathbb{1}_{S|B}. \tag{85b}$$

## Compatibility with an external zero-charge state

Here we prove that, for compact $G$, any state $\rho_{\overline{S|A},S|A}$ in Alice's perspective is compatible with the existence of a reference frame B and a perspective-neutral pure state $|\phi\rangle_{ABS}$. Without loss of generality, we restrict Eq. (49) to a pure state, so that

$$\rho_{\overline{S|A},S|A} = \int dg\,dh\, R_{\overline{S|A}}(g^{-1}h) \otimes U_{S|A}^\dagger(g)|\psi(g)\rangle\langle\psi(h)|U_{S|A}(h). \tag{86}$$

Consider now, in the standard partition, the zero-charge state $|\phi\rangle_{ABS}$ defined by

$$|\phi\rangle_{ABS} = \int df\,dg\,dh\,|fg\rangle_A \otimes |fh\rangle_B \otimes U_S(f)|\phi(g,h)\rangle_S, \tag{87}$$

where

$$|\phi(g,h)\rangle_S = \frac{1}{|G|} U_S(h)|\psi(h^{-1}g)\rangle_S. \tag{88}$$

Tracing out B from $|\phi\rangle_{ABS}$ and then changing to Alice's perspective via the $\mathcal{E}_A$ map defined in the main text, we obtain $\rho_{\overline{S|A},S|A}$, as we wanted to show.

## Data availability
Data sharing not applicable to this article as no datasets were generated or analysed during the current study.

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

## Acknowledgements

We thank T. Galley and P. Hoehn for comments on an earlier version of this draft. We thank C. Rovelli for discussions. This publication was made possible through the support of the ID# 61466 grant and ID# 62312 grant from the John Templeton Foundation, as part of the project 'The Quantum Information Structure of Spacetime' (QISS). The opinions expressed in this publication are those of the authors and do not necessarily reflect the views of the John Templeton Foundation. This work was supported by the Program of Concerted Research Actions (ARC) of the Université libre de Bruxelles. O.O. is a Research Associate of the Fonds de la Recherche Scientifique (F.R.S.-FNRS). E.C-R. acknowledges financial support from the Austrian Science Fund (FWF) through BeyondC (F7103-N48), the European Commission via Testing the Large-Scale Limit of Quantum Mechanics (TEQ) (No. 766900) project, the Foundational Questions Institute (FQXi), the Swiss National Science Foundation (SNSF) via the National Centers of Competence in Research QSIT and SwissMAP, as well as the project No. 200021_188541. O.O. acknowledges support from the F.R.S.- FNRS under project CHEQS within the Excellence of Science (EOS) program.

## Author contributions

E.C.-R. and O.O. contributed equally to this work.

## Competing interests

The authors declare no competing interests.
