## [Transparent Peer Review file · Communications Physics]

Relative subsystems and quantum reference frame transformations

Corresponding Author: Dr Esteban Castro Ruiz

This manuscript has been previously reviewed at another journal that is not operating a transparent peer review scheme. The manuscript was considered suitable for publication with further review at Communications Physics. This document only contains information relating to versions considered at Communications Physics.

Version 1:

Reviewer comments:

Reviewer #2

(Remarks to the Author)

The authors have kindly clarified the points I raised to my satisfaction and have described the changes in detail in the accompanying letter. However, one remaining point of disagreement pertains to the discussion on page 18. I respectfully disagree with the authors' assertion that "There is enough physical insight and similarity to the case of translations to make our results compelling." I also question the rationale behind the claims being "compelling" based on "physical insight" as a standard of knowledge. Even though I maintain that my objections are valid, I do see that this paper makes a contribution to the field which others will find interesting and important, and therefore I think it is reasonable to publish it.

Reviewer #3

(Remarks to the Author)

The paper presents a novel operationally-defined approach to quantum reference frames (QRFs) and utilize it to inquire about the QRF-dependence of the notion of subsystems defined relative to a QRF. Over the years, QRFs have gained an increasing role in various fields and the ensuing quantum frame relativity of subsystems has been recently identified as a core feature of various approaches to QRFs with several physically relevant implications, ranging from quantum foundations and quantum information to gauge theories and gravity. A better understanding of the foundations and assumptions underlying the various QRF approaches is thus of key importance not only for those who already work on this topic but also for other research communities. Given this situation, the topics discussed in the paper are of broad relevance and, in my opinion, the manuscript is well written and fairly accessible to both QRF experts and outsider readers. In particular, I appreciated the effort made by the authors in comparing their approach with others, with an emphasis on the main differences.

The authors' framework is based on incoherent rather than coherent group averaging and this constitutes a major difference compared to some other approaches to QRFs, both at a conceptual and technical level. Building up on this starting point, the authors claim to solve some of the "problematic" features of other approaches by yielding reversible QRF-transformations that only depend on the reference frames and system of interest. Despite of the additions already made by the authors to address the comments of Referee 1, I find that further motivations for the authors' starting point and further comparison with existing literature would still be required.

Some more specific and detailed comments are below.

- About incoherent vs. coherent group averaging and external-frame independence:

In Sec. I and II, the authors argue that observers who lack access to the external reference frame are constrained to density operators ρ that are invariant under the G -action, namely $U(g)\rho U(g)^\dagger = \rho$, for any $g \in G$, a weaker requirement than demanding invariance $U(g)|\psi\rangle = |\psi\rangle$ of pure states $|\psi\rangle$ under G . This amounts to consider the larger algebra A_{inv} of invariant operators rather than its subalgebra A_{phys} of gauge-invariant/external-frame-independent operators in the sense of Dirac constraint quantization as it is instead the case in the perspective-neutral approach to QRFs. However, there are arguments in the literature (see e.g. Sec. II.B3 in quant-ph/2308.09131) according to which invariant

states are not fully external-frame independent in that the incoherent group averaging projector distinguishes for example between a superposition and a mixture of gauge-related/internally indistinguishable states. Full external-frame independence is instead achieved on the physical Hilbert space of perspective-neutral states and it is only when restricted to the latter that invariant and physical operators agree. I believe it would then be worth for the authors to expand on their motivation in favour of invariant states/operators and explicitly compare the arguments they give in Sec. II with the above. Clearly, the use of invariant states is not invalidated in operational situations where some external frame information is permitted (e.g. the Lab). If this is what the authors have in mind, it should be clarified. In particular, given the authors' emphasis on say Alice's apparatus not being part of the quantum system under consideration in Sec. III, it should be clarified whether and how this relates to the statement that observers actively use a quantum reference frame to make measurements.

- About the extra particle:

A key point of the paper is the refactorization in Sec. IV.B of the starting external-frame-dependent total Hilbert space $H_A \otimes H_B \otimes H_S$ as $H_C \otimes H_{\{BS|A\}}$ where H_C decomposes into a left- and a right-invariant part. This reflects in Sec. IV.C into a factorization of the total invariant algebra into the $\overline{BS|A}$ and $BS|A$ factors, the former being identified with the left-invariant part of C . The inclusion of $\overline{BS|A}$, interpreted as the "extra particle", is crucial for the unitary transformations constructed in Sec. V as it plays the role of a purifying party. In light of my previous comment about the invariant algebra, I am wondering whether the inclusion of the extra particle, which emerges as a consequence of the invariant degrees of freedom of the reference frame, is a reflection of some remaining redundant information about the external frame which is not to be gauged away at the level of the invariant algebra. Still, if this is the case, the conclusions about the frame-dependence of relative subsystems should not be affected as the perspectival degrees of freedom of say S relative to the chosen frame should be equivalent, at least for ideal/perfect frames, to those identified passing through the smaller purely external-frame-independent physical algebra obtained via coherent group averaging. From a conceptual point of view, it is perhaps again worth to clarify these aspects though.

- About the discussion:

Both in the introduction and in the discussion section, the authors explicitly mention quantum gravity among their motivations, at least at a broad level, and the importance of QRFs for it. In this respect, to address and clarify the previous comments becomes quite relevant. More specifically, in the last paragraph of p. 25, it is explicitly mentioned that the framework developed in the paper might be a good starting point towards incorporating general relativistic symmetries, which would be important in quantum gravity. Now, I of course agree that a better understanding of QRF transformations is a very important step in this direction but I find myself confused by the authors' will/hope to pursue this by favouring a incoherent over a coherent twirling. Since in a fully constrained system like gravity one would impose diffeomorphism-invariance in the stronger sense of Dirac quantization, how does this reconcile with the authors' main point? In particular, a few lines later at the beginning of p. 26, they also mention the need to go beyond regular representations and perfect frames. But as discussed in the Ref. [80] cited by the author (see also quant-ph/2308.09131), for non-ideal frames, the quantum information-theoretic approaches to QRFs are not equivalent to the perspective-neutral approach based on the stronger notion of invariance as in constraint quantization. A framework based on the weaker notion of invariance as the one used by the authors is thus expected to be leading to inequivalent conclusions beyond ideal frames.

Minor comment:

In the second to last paragraph of p.5 it is written "...the jumping rules of Refs. [37, 39] could be valid for any reference frames A and B only in two circumstances:..." but then only circumstance 1) is mentioned. Is it a typo or am I missing something?

Version 2:

Reviewer comments:

Reviewer #3

(Remarks to the Author)

I thank the authors for their detailed reply and the numerous clarifications added to the manuscript. However, I respectfully disagree with two points in the authors's reply.

The first is the statement that the imposition of the constraints in the sense of Dirac quantization, and hence the need of a coherent group averaging, would not be required by symmetry considerations alone and would rather require to further impose a charge sector. The second strictly related point is again about external frame independence. Let consider for example a system of N particles in 1 dimension. We can think of the absolute positions of the particles (the kinematical degrees of freedom) as being specified relative to the position of some other particle external to the system playing the role of an external frame and whose position is set to be say in the origin. Now, wiping out any reference to the external frame amounts to require the total momentum of the particles to be vanishing. Gauge transformations, which amount to a translation of the N particles, can in fact be equivalently thought of as a translation of the external frame and gauge-invariant states are those annihilated by the constraint. This does not necessarily require to invoke arguments about a Lagrangian and dynamics. Clearly, it would also be true if one starts with a translation-invariant Lagrangian but the above considerations holds only on the base of requiring gauge-invariance/external frame-independence. A purely internal description of the system of particles then requires not only the observables to be gauge-invariant but also the states to satisfy the constraints.

The gauge-invariant states are obtained via coherent group averaging and, upon completion with respect to a suitable inner product, form the physical Hilbert space of Dirac quantization. For this reason, I think that a superposition and mixture of gauge-related states should not be distinguishable without having access to the external frame (and this is not based on classical intuition). I therefore think of the fact that these states can be distinguished by operations in the invariant algebra considered by the authors as signalling that some remnant of the external frame information is still present if one only requires invariance of observables under unitary conjugation without considering what is happening at the level of the states. The considerations in Appendix A are not in contrast with this as one can easily convince himself that, for any kinematical operator A , the following relation holds true $\Pi A \Pi = G(A)\Pi$, where Π denotes the coherent group averaging projector and $G(A)$ the G -twirl, i.e., coherent and incoherent group averaging agree on the physical Hilbert space. A fully external frame-independent description should however be such both at the level of observables and states and is achieved on H_{phys} .

This being said, I think the paper represents an interesting contribution to the QRF literature and the authors addressed most of my previous comments and clarified their motivations to a sufficient degree to make the manuscript reasonable to publish. I would nevertheless appreciate to hear the authors' opinion on the above discussion and, in case they agree, modify/adjust accordingly the comments added on page 5 and 10 of the revised manuscript.

We thank the Reviewers for their comments. Below, we give a point-by-point response to their remarks. We also point out the changes that we have made to our manuscript in view of the Reviewer's observations.

Reviewer #1 (Remarks to the Author):

Summarize:

The authors argue that standard approaches to quantum reference frames (QRFs) are problematic because they require that all degrees of freedom in the theory are included in transformations between frames, leading to an irreversibility if only some of them are included. They present an approach to QRFs based on incoherent averaging over the symmetry group for which the QRF provides a reference. This loosens the requirement that the conserved charge associated with the symmetry group is well-defined, and results in the emergence of additional degrees of freedom which the authors refer to as an "extra particle".

The manuscript is an interesting contribution to the literature, with a mathematical rigor that is sometimes lacking in the field. Furthermore, the authors' attempts to make the abstract formalism of QRFs more operational is desirable for the field as a whole. However, I do not find the authors' argument that standard QRF transformations are problematic convincing (see below). Without any problematic physical predictions, this apparent problem seems to just be a matter of taste. Accepting the premise anyway, the authors' solution seems to come at the cost of the usual statistical interpretation of the density matrix (see below), as well as the necessary existence of a "extra particle". The latter cost is quite high, given that the authors' goal was to develop an approach which only required to include a specific subset of the degrees of freedom present in the theory. However, I am sure the QRF community will find some value in considering the authors' incoherent averaging approach.

Some more specific and detailed comments are below.

About the "problem" identified by the authors:

The example given Sec. II (page 4), describes the well-known situation where quantum correlation is created by transforming from a perspective where a given particle is in superposition with respect to some system, to the perspective of that superposed particle itself. This is presented as problematic due to the potential presence of some extra system which would likewise become correlated, resulting in an impure state for the initially considered systems. However, according to standard approaches to QRFs, this is neither unexpected nor problematic. Therefore whether or not this is a "bug" or a "feature" is a question of taste, as I will argue in the following.

On this topic, the authors say in the introduction that:

"reversible transformations between the descriptions relative to different arbitrary QRFs are in general obtained only when these descriptions include the whole rest of the universe. This "nonlocality" of the prescriptions is unsettling from a conceptual

point of view as it goes contrary to the intuition that predictions concerning local systems should require only local data, raising the question of whether a local approach could be developed”

While there is some value to examining an entirely ”local” (in the sense quoted above) approach and comparing it with standard approaches, I do not see the problem with the fact that in standard approaches, a change of reference frame can affect every object in the theory. This also occurs classically: for example, a classical translation of a coordinate system will change the coordinates of everything in the theory, including things arbitrarily far away. If we perform effectively a superposition of these translations (as in QRFs), it is to be expected that those arbitrarily far away things become entangled.

We respectfully disagree with this comment. The problem we discuss in Section 2 is not simply ”a question of taste”. Rather, it leads to different physical predictions, depending on whether the extra system is included or not. To emphasise this point, we rewrote the last paragraph of page 4. As we point out, adding the extra system from A’s perspective means that the state of AS relative to B is mixed, whereas not doing it means that the state is pure. This clearly implies different physical predictions, as a mixed state can always be distinguished from a mixed one by doing sortable measurements.

Invoking translations in classical physics, Reviewer #1 argues that entanglement in the quantum case is to be expected. However, for the reasons explained above, this makes the quantum case completely different to the classical one. The problem we point out is clearly a serious one for QRF transformations. In this work, we show how our new framework can solve it completely.

About the authors’ operational justification:

The authors justify their perspective by saying that ”in this paper we are interested in the case where observers actively use a quantum reference frame to make measurements on a system and not on mere changes of coordinates”. I appreciate the authors’ desire to put the formalism of QRFs on a more operational foundation, as this is often missing from the QRF literature in my opinion. Unfortunately it is not defined what it means to ”actively use a quantum reference frame to make measurements on the system”, so I don’t see how this is more physical than ”mere changes of coordinates”.

We thank Reviewer #1 for this comment. To clarify the interpretational difference between our framework and ”mere changes of coordinates”, we modified the end of the first paragraph on page 6. We point out that our framework describes transformations between ”the invariant, operationally-defined description of one observer, who can only perform invariant measurements on a subsystem of a potentially greater system, to the invariant, operationally-defined description of another observer with the same restrictions. ” This interpretation is different from the one in which QRF transformations are seen as (quantum) changes of coordinates. We point this out in footnote 2, where we contrast our findings with the operational interpretation of the ”perspective neutral” approach offered in reference 50.

About the incoherent group average:

By making the weaker assumption of symmetry of the density matrix instead of symmetry of the pure states, we lose the statistical-prep interpretation of the density matrix for the following reason. In general, such a density matrix can be decomposed into a mixture of pure states which themselves aren't symmetric, and therefore aren't allowable on physical grounds. This is fine on average, as the transformations of the constituent states act to counteract each other in the statistical ensemble, but it would allow the symmetry to be broken on a single-shot experiment. The authors' formalism therefore requires that the statistical interpretation of the density matrix has to be given up.

We respectfully disagree with this comment. It is a fundamental fact of quantum mechanics that no experiment can tell the difference between two different mixtures of pure states that lead to the same different density operator. For this reason, it is impossible to break the symmetry in an experiment (either single- or multiple-shot), as the Referee suggests.

About the conclusion:

The authors imply in their concluding paragraph that their approach may make it possible to study QRFs with respect to the Lorentz and Poincaré groups, though they don't make it clear why their approach should be any more suited to this than standard approaches.

We thank reviewer #1 for this comment. In the last paragraph of the Discussion, we added the following: "These QRF transformations those for [Lorentz and Poincaré groups] would allow us to study proper subsystems transforming under Lorentz and Poincaré symmetries, which is not thoroughly understood, and would be an important step towards ultimately incorporating general relativistic symmetries, which would be important in quantum gravity."

Reviewer #2 (Remarks to the Author):

The manuscript provides a novel approach to quantum reference frames and their transformations, arguably generalizing other recent attempts. It differs conceptually and formally and provides interesting new insights into the matter. The formalism is aligned with the incoherent approach to group averaging, and relational in a strong sense. The authors point out the need for including in the description of a frame transformation an 'extra particle', whose state carries the missing global information that is necessary for the unitarity of the frame change procedure. The formalism uses algebraic language, describing subsystems primarily as subalgebras. It is written in a relatively accessible way and cites the most relevant sources.

The work is valid and valuable as a conceptual and heuristic input to the intensely debated problems. The points raised and discussed are a significant contribution to the theory of quantum reference frames that is still under development. It should not, however, be seen as providing a rigorous foundation for quantum reference frames and frame changes. This is because, unfortunately, it contains unwarranted

claims with no proofs, which moreover I believe are not correct in the form they are stated. I will list these points here.

We thank Reviewer #2 for noticing and appreciating the conceptual contribution of our work.

Before proceeding with a detailed reply to the Reviewer's comments, we would like to start with a general remark. The main focus of our work is conceptual. As such, our work is not, and is not meant to be, at the level of mathematical rigour found in mathematical physics journals. Nevertheless, our work is very rigorous from the physics point of view. We take the interpretation of QRF transformations very seriously, as the Reviewer notices. We think that our paper sheds important light on some of the deepest conceptual issues regarding QRFs, an achievement that deserves high visibility. In our opinion, the most pressing problems regarding QRFs are conceptual in nature. This is why we decided to put our efforts mostly in this direction. Having said this, mathematical rigour is of course very important as well. We therefore thank Reviewer #2 for the comments in this regard, which have greatly helped us to improve the quality of our work.

- pg. 7, bottom: after eq. (5) it is mentioned that such representation of states is correct and valid "more generally" than for compact groups. The only source cited treats only compact groups, and it is unclear from the discussion how to deal in this way with any infinite-dimensional representation, that is necessary encountered in the context of non-compact groups. Distributional techniques could be implicit here, but if that was the intention, it was never mentioned and implementation of this strategy is not known for general groups. Besides these facts, the framework is declared to hold for arbitrary unimodular groups (pg. 2). Sadly, these issues flaws the whole consequent analysis, making the contribution heuristic.

We thank the reviewer for this comment. As we mention in the third paragraph of page 8 (current version), Eq. 5 reduces to the well-known expression of the Fourier transform when G is the translation group. Although strictly speaking the resulting vectors $|x\rangle$ are not normalisable, their use is standard in the physics literature. If handled properly (i.e. along with suitably normalised wave packets), they are very useful and always lead to correct conclusions. Therefore, the translation group is a concrete example where Eq. 5 applies more generally, not only for compact groups. Mathematically, our treatment of the (centrally extended) 1d Galilei group is not so different to the case of the translation group, so we believe our work is mathematically sound. We have rewritten parts of the discussion below Eq. 5 to make these points clearer.

- pg. 12, top: I see no justification for the existence of the tensor product decomposition of eq. (10). This is also crucial for the subsequent analysis.

The existence of such a decomposition can be understood in terms of the isomorphism defined below Eq. 11. We have modified the text below Eq. 10 to make this point clear.

- pg. 12, middle: the understanding of the formula (12) as defining a unitary operator on a separable Hilbert space only makes sense if G is countable.

Moreover, the claims coming just after (12) are presented as following from this unwarranted claim, but I fail to see how.

The unitarity of the operator defined in Eq. 12 is a consequence of the orthogonality of the vectors $|g\rangle$. We modified the text below Eq. 12 to clarify this point. (The same comment as above applies here as well: although not normalisable, the vectors $|g\rangle$ are widely accepted in the physics literature and very helpful if handled with care.)

- pg. 18: as admitted by the authors, the analysis of the QRFs for the extended Galilean group is not done rigorously; the cited work uses entirely different mathematical tools. Thus it should be treated with caution, more as containing heuristics than results.

We respectfully disagree with Reviewer #2 here. There is enough physical insight and similarity to the case of translations to make our results compelling. Regarding reference 10, we thank the Reviewer for this remark. To improve our presentation in this regard, at the end of the introduction to Section 4, we have included a sentence that reads: "It would be interesting to see if our construction can be cast in the rigorous formulation of covariant "screen observables" for the Galilei and Poincaré groups [68]".

Let me also point out some minor nitpicks:

- pg. 10, bottom: $B_{\text{inv}}(H_{AS}|E)$ is not a Hilbert space, but a Banach space.

We thank Reviewer #1 for this remark. We have changed the bottom of page 10 so that it reads "vector space" instead of "Hilbert space".

- pg. 11, top: It is not entirely clear where the operators T and $T_S|A$ act

We thank Reviewer #2 for this comment. The discussion before Eq. 8 is a motivation for the definition of the relative observables $T_{S|A}$. Therefore, the definition of "operator T with respect to A " is not made precise yet. This is made on purpose to present the ideas in a physically intuitive manner. To clarify where the operators act, we have added the subscript $S|A$ above Eq. 8 and in Eq. 8.

- pg.11, after eq. (8): I disagree $S|A$ to be independent of the tensor product decomposition - it is defined with it, by specifying the frame (with the regular representation). Or I am confused here, maybe this point just needs clarification of what exactly is meant.

We thank Reviewer #2 for this comment. Abstractly, the algebra $S|A$ can have different representations in different tensor product decompositions. This is what we mean by the algebra being independent of the tensor product decomposition. We have modified the text below Eq. 8 to clarify this point.

- the tension with the perspective-neutral approach to unitary frame changes is mentioned a couple of times, but not explained: if the frame transformations presented there are not unitary, perhaps the authors should point out where concretely mistakes are made in deriving an opposite conclusion.

There is no mistake in the “perspective neutral” approach. The difference between our framework and the perspective neutral one is the assumption of a vanishing total charge. We believe our changes to Section 2, in view of the comments by Reviewer #1, sufficiently clarify the difference between the approaches

Since I find the ideas presented in the manuscript highly valuable, I refrain from advising against publication; unfortunately, the presented level of mathematical rigour and clarity is ubiquitous in the literature on the subject, so perhaps the editor will not find my concerns worrying. Instead, I suggest tools that may help provide a rigorous treatment of some of the presented intuitions.

The relative operators seem to be defined to be of the form of eq. (8), and frames are assumed to carry a regular representation of the group. This is precisely the setup of relative observables for ideal frames as defined in arXiv:2303.14002. There it is also proved that such operators indeed form a subalgebra of the invariant algebra of the composite system. The question if the commutant of the relative algebra, which is the 'gauge system', complements it to the full invariant algebra could be properly addressed in this language. From here it is still far to the decomposition in terms of Hilbert spaces, but perhaps the physical claims could be made on the algebraic level, which seems to be the authors' preference.

Jan Głowacki

We thank again the Reviewer for the positive comments and for the useful reference. We have included it in to our bibliography, with a reference to it in the introduction to Section 4.

Reviewers' comments:

Reviewer #2 (Remarks to the Author):

The authors have kindly clarified the points I raised to my satisfaction and have described the changes in detail in the accompanying letter. However, one remaining point of disagreement pertains to the discussion on page 18. I respectfully disagree with the authors' assertion that "There is enough physical insight and similarity to the case of translations to make our results compelling." I also question the rationale behind the claims being "compelling" based on "physical insight" as a standard of knowledge. Even though I maintain that my objections are valid, I do see that this paper makes a contribution to the field which others will find interesting and important, and therefore I think it is reasonable to publish it.

Authors' response (AR):

We thank the reviewer for their positive remarks. Regarding their objection to our formulation of Galilean quantum reference frames, we believe that we have clearly acknowledged that this is not an entirely rigorous treatment for the reasons raised by the referee, which we have described in the manuscript. Having said this, we also think that the results we have derived, which are described in terms of linear combinations of position and momentum operators in analogy with the classical case, are particularly elegant and offer a quite significant understanding about the essential principles that govern the physics of non-relativistic particles relative to a quantum reference frame.

Change to the manuscript: To improve our presentation, we have rewritten the expression for the relative generators of the Galilei group in Eqs. 28. Below this equation, we have added a short remark on the physical interpretation of the integral terms in these equations.

Reviewer #3 (Remarks to the Author):

The paper presents a novel operationally-defined approach to quantum reference frames (QRFs) and utilize it to inquire about the QRF-dependence of the notion of subsystems defined relative to a QRF. Over the years, QRFs have gained an increasing role in various fields and the ensuing quantum frame relativity of subsystems has been recently identified as a core feature of various approaches to QRFs with several physically relevant implications, ranging from quantum foundations and quantum information to gauge theories and gravity. A better understanding of the foundations and assumptions underlying the various QRF approaches is thus of key importance not only for those who already work on this topic but also for other research communities. Given this situation, the topics discussed in the paper are of broad relevance and, in my opinion, the manuscript is well written and fairly accessible to both QRF experts and outsider readers. In particular, I appreciated the effort made by the authors in comparing their approach with others, with an emphasis on the main differences.

AR:

We thank the Reviewer for these positive remarks and appreciate that they found our work well written. We thank them for their critical comments too, as they helped us make our point clearer.

The authors' framework is based on incoherent rather than coherent group averaging and this constitutes a major difference compared to some other approaches to QRFs, both at a conceptual and technical level. Building up on this starting point, the authors claim to solve some of the "problematic" features of other approaches by yielding reversible QRF-transformations that only depend on the reference frames and system of interest. Despite of the additions already made by the authors to address the comments of Referee I, I find that further motivations for the authors' starting point and further comparison with existing literature would still be required.

AR:

We thank the Reviewer for this general remark. In section II, we showed that a theory of quantum reference frame transformations based on coherent twirling has problems when considering

potential external systems that are not invariant under the action of the group. A theory based on incoherent twirling is more suitable to accommodate these situations, which are common to almost all situations in physics. Although we believe that compatibility with potential external frames is a strong reason to develop a theory of quantum reference frame transformations based on incoherent twirling, we take the opportunity to further expand on our motivations and assumptions.

Let us consider a simple scenario. Suppose an observer measures relative quantities of a system with respect to their reference frame. Including the reference frame into our description, these measurements are represented by operators given by Eq. 8 of our manuscript. Now suppose the observer loses access to this reference frame. What quantities can the observer measure now? Clearly, the answer is given by those operators of the form in Eq. 8, which are localised on the system alone, that is, which are of the form $\mathbb{1}_A \otimes T_S$. We recognise immediately that the most general quantities the observer can measure are the (incoherently twirled) invariant observables. Therefore, a symmetry constraint, in the sense of a lack of access to a reference frame, restricts our measurements to the set of invariant operators *and not further*.

Importantly, the set of invariant operators is reference frame independent in a strong sense: as we prove in appendix A, for any two observers (Eve and Fer, say) who possess quantum reference frames E and F , respectively, and for any system Q , the invariant sub algebra $L_{\text{inv}}(H_Q)$ is the largest sub algebra common to the algebras of relative observables (as defined by Eq. 8) with respect to E and F . This implies that $L_{\text{inv}}(H_Q)$ is independent of any potential external frame.

As a consequence of this analysis, we see that symmetry considerations alone do not imply the coherent twirling approach. Indeed, when we use coherent twirling instead of incoherent, we do more than implementing a symmetry resulting from the lack of a reference frame — we impose a charge sector. In our view, a fully satisfactory theory of quantum reference frame transformations should be free from this restriction and based on symmetry considerations alone.

Change to the manuscript: In the previous version of our manuscript, the argument that leads to the invariant sub algebra from losing access to a reference frame was inserted as a footnote. In the revised version, we have moved it to the main text (second paragraph of page 10). Moreover, we have included the statement that symmetry considerations alone do not imply a coherent group averaging approach. We believe this change further clarifies our motivation.

Some more specific and detailed comments are below.

- About incoherent vs. coherent group averaging and external-frame independence:

In Sec. I and II, the authors argue that observers who lack access to the external reference frame are constrained to density operators ρ that are invariant under the G -action, namely $U(g)\rho U(g)^\dagger = \rho$, for any $g \in G$, a weaker requirement than demanding invariance $U(g)|\psi\rangle = |\psi\rangle$ of pure states $|\psi\rangle$ under G . This amounts to consider the larger algebra A_{inv} of invariant operators rather than its subalgebra A_{phys} of gauge-invariant/external-frame-independent operators in the sense of Dirac constraint quantization as it is instead the case in the perspective-neutral approach to QRFs. However, there are arguments in the literature (see e.g. Sec. II.B3 in quant-ph/2308.09131) according to which invariant states are not fully external-frame independent in that the incoherent group averaging projector distinguishes for example between a superposition and a mixture of gauge-related/internally indistinguishable states. Full external-frame independence is instead achieved on the physical Hilbert space of perspective-neutral states and it is only when restricted to the latter that invariant and physical operators agree. I believe it would then be worth for the authors to expand on their motivation in favour of invariant states/operators and explicitly compare the arguments they give in Sec. II with the

above. Clearly, the use of invariant states is not invalidated in operational situations where some external frame information is permitted (e.g. the Lab).

If this is what the authors have in mind, it should be clarified. In particular, given the authors' emphasis on say Alice's apparatus not being part of the quantum system under consideration in Sec. III, it should be clarified whether and how this relates to the statement that observers actively use a quantum reference frame to make measurements.

AR:

We thank the Reviewer for pointing us to the argument in quant-ph/2308.09131. In the following we comment on several aspects of this argument and the Reviewers' remark.

Methodology:

The reasoning behind Sec II B in quant-ph/2308.09131 is based on an analogy between the perspective-neutral framework and special relativity (SR). The authors argue that the perspective-neutral framework is the most compatible one with reference frame changes in SR. Methodologically, our work differs from the motivation presented there. In our view, a completely general framework for quantum reference frame transformations should not be constructed trying to accommodate the paradigm of SR (or any other classical theory for that matter).

External reference frame independence:

In particular, a criterion for "external frame independence" should not be based, in our opinion, on an analogy with a classical theory. Furthermore, as we have shown, symmetry considerations alone do not imply zero global charge. Rather, external frame independence should be understood on the basis of operational principles like the ones we present in our appendix A. By construction, our framework is completely external frame independent. Yet, it is perfectly consistent with the potential existence of external frames, something that is problematic if one fixes only to the zero-charge subspace, as we show in section II. The existence of new degrees of freedom — the extra particle — are key to this consistency. In our view, a general theory of quantum reference frame transformations should embrace the existence of these degrees of freedom. The extra particle is a direct consequence of imposing a symmetry constraint in the most general terms — and it is important.

Superposition vs mixture:

Our framework distinguishes between the two situations mentioned by the Reviewer because these situations are indeed different, and the difference can be checked by means of invariant operations. In our view, the expectation that they are both the same comes from a classical intuition that does not apply to quantum superpositions. As we have argued above, these operations are external frame independent in the sense of appendix A. An important precision, however, is in order: the state of the system relative to the reference frame, $S|A$, does not distinguish between a superposition and a mixture of a state and its "rotated version". However, the extra particle $\overline{S|A}$ does distinguish between these two situations. As we mention in the abstract, these degrees of freedom characterise the quantum features of reference frame states.

About conditioning:

In the perspective neutral framework there are two equivalent ways of "jumping" to the perspective of an internal quantum reference frame. One can apply a unitary that "disentangles" the quantum reference frame degrees of freedom and then ignore them — they are redundant. Alternatively one can condition on the reference frame being at the origin (the state corresponding to the identity element of the group). In our framework, conditioning on the reference frame being at the origin

gives the state of the system relative to the frame, that is the state on $S|A$, just like in the perspective-neutral framework (the actual states one gets are different, of course). However, in our framework, the state on $S|A$ is not the complete invariant information, as the subsystem $\overline{S|A}$ has to be taken into account. For this reason, we define "jumping" to the perspective of an internal quantum system by means of the operator $V_{\{E \rightarrow A\}}$, defined in Eq. 12.

Changes to the manuscript: We have added a discussion about the coherent superposition vs probabilistic mixture point raised by the Referee at the end of Section IV. We have cited the reference mentioned by the Reviewer and contrasted our results to those of the perspective neutral approach.

- About the extra particle:

A key point of the paper is the refactorization in Sec. IV.B of the starting external-frame-dependent total Hilbert space $H_A \otimes H_B \otimes H_S$ as $H_C \otimes H_{\{BS|A\}}$ where H_C decomposes into a left- and a right-invariant part. This reflects in Sec. IV.C into a factorization of the total invariant algebra into the $\overline{BS|A}$ and $BS|A$ factors, the former being identified with the left-invariant part of C . The inclusion of $\overline{BS|A}$, interpreted as the "extra particle", is crucial for the unitary transformations constructed in Sec. V as it plays the role of a purifying party. In light of my previous comment about the invariant algebra, I am wondering whether the inclusion of the extra particle, which emerges as a consequence of the invariant degrees of freedom of the reference frame, is a reflection of some remaining redundant information about the external frame which is not to be gauged away at the level of the invariant algebra. Still, if this is the case, the conclusions about the frame-dependence of relative subsystems should not be affected as the perspectival degrees of freedom of say S relative to the chosen frame should be equivalent, at least for ideal/perfect frames, to those identified passing through the smaller purely eternal-frame-independent physical algebra obtained via coherent group averaging. From a conceptual point of view, it is perhaps again worth to clarify these aspects though.

AR:

We thank the Reviewer for this comment. We believe that they have an interesting intuition about the extra particle, which we now try to formulate in a precise way. First, the extra particle represents true invariant degrees of freedom. As such, it is not "the reflection" of redundant degrees of freedom and cannot be "gauged away". However, there are at least two important ways in which the extra particle allows our formalism to be "consistent with the potential existence of external reference frames" outside of our description. We now explain in detail each of them.

On the one hand, given any state $\rho_{\{\overline{S|A}, S|A\}}$ of the extra particle and the system "as seen" from Alice, there always exists a reference frame B (as seen from the perspective of the external observer, Eve) and a zero-charge pure state $|\phi\rangle_{\{ABS\}}$ such that tracing B out and then "jumping" to Alice's perspective recovers $\rho_{\{\overline{S|A}, S|A\}}$. We believe this fact constitutes an interesting link between our framework and the perspective-neutral formalism: roughly speaking, the extra particle allows for a "purification" of our state in the zero charge subspace of a larger system (from the perspective of Eve). We would emphasise, however, that our framework is formulated in terms of purely invariant degrees of freedom, so such an extension to a perspective neutral state is possible but by no means necessary. In fact, extending to the zero-charge subspace means adding more variables (gauge and invariant) than strictly necessary.

On the other hand, the extra particle is what allows the formalism to work based on the same principles both for the internal and the external description. If we ask how some external frame, say B , looks in the perspective of A , this is a subsystem $B|A$ that is not separate from the extra particle

$\bar{S|A}$ arising in Alice's description of S . In fact, this is true for any system other than S that Alice may bring to her description (of course, if we define the new extra particle $\bar{S|B|A}$, it will be separate from both $S|A$ and $B|A$; note also that relative systems, such as $S|A$, $B|A$, $C|A$, etc., are always separate). In this sense, we may say that the extra particle arising in the description of a specific system S contains information about any other system, such as the frame B , that may be considered. This does not, however, mean that the extra particle $\bar{S|A}$ contains any information obtained by using B as a reference frame.

Changes to the manuscript: We have added, in the second-to-last paragraph of page 14, a comment stating that the extra particle is important for the consistency of our framework with the potential existence of external frames.

We have also discussed, on page 18, that the extra particle $\overline{S|A}$ arising in the description of a given system S by a given frame A overlaps with any relative system, such as $B|A$, that may be brought into the description relative to A , and in this sense can be said to contain information about the 'rest of the world' relative to A . This explains, for example, why if we jump from a classical frame to a frame in a superposition, any system that was previously in a pure noninvariant state would look correlated with any other such system in the universe, yet its purification can be found in its corresponding extra particle without violating the monogamy of entanglement. We have also emphasised that, nevertheless, the extra particle does not contain gauge degrees of freedom as it is fully within the invariant subsystem of the system and frame.

On page 25, we have added a paragraph stating that our framework is consistent with an external reference frame and a perspective-neutral state. However, importantly, we have also emphasised that this extension is not at all necessary for the interpretation of our formalism. The proof of our claim is included in a new appendix (Appendix I).

- About the discussion:

Both in the introduction and in the discussion section, the authors explicitly mention quantum gravity among their motivations, at least at a broad level, and the importance of QRFs for it. In this respect, to address and clarify the previous comments becomes quite relevant. More specifically, in the last paragraph of p. 25, it is explicitly mentioned that the framework developed in the paper might be a good starting point towards incorporating general relativistic symmetries, which would be important in quantum gravity. Now, I of course agree that a better understanding of QRF transformations is a very important step in this direction but I find myself confused by the authors' will/hope to pursue this by favouring an incoherent over a coherent twirling. Since in a fully constrained system like gravity one would impose diffeomorphism-invariance in the stronger sense of Dirac quantization, how does this reconcile with the authors' main point? In particular, a few lines later at the beginning of p. 26, they also mention the need to go beyond regular representations and perfect frames. But as discussed in the Ref. [80] cited by the author (see also quant-ph/2308.09131), for non-ideal frames, the quantum information-theoretic approaches to QRFs are not equivalent to the perspective-neutral approach based on the stronger notion of invariance as in constraint quantization. A framework based on the weaker notion of invariance as the one used by the authors is thus expected to be leading to inequivalent conclusions beyond ideal frames.

AR:

We would like to emphasise again the point that a lack of reference frame for a given group does not imply zero charge, as the general argument given above shows. In our view, Dirac quantisation is not simply a prescription of how to express a gauge symmetry in quantum mechanics, but it is about quantising a full dynamical theory, given by a Lagrangian, which has such a symmetry.

Understanding the precise connection between our results and the principles underpinning Dirac quantisation is a very important question in our view, but it goes beyond the scope of the present work. We see our paper as first step in this direction, which investigates the consequences of pure symmetry considerations from an operational perspective. It is our hope that this will trigger further investigations into this question and ultimately provide deeper understanding of the principles underpinning the quantisation of systems with gauge symmetries. We note that there are examples in the quantum gravity literature that do not impose a total charge equal to zero in the Hamiltonian constraint: <https://arxiv.org/abs/2302.06603> .

Something that we can certainly say, however, is that our framework is compatible with a quantum system subject to the zero-charge constraint, if we regard this constraint as an additional, physical condition.

Regarding the comment of the Reviewer about the application of our formalism to non-ideal frames and its relation to the perspective neutral framework, we believe that a proper comparison should be based on a concrete physical situation where both frameworks apply. The frameworks need not differ a priori. If the total charge vanishes, both frameworks coincide and they will give equal predictions. If the total charge does not vanish, then we will have a situation outside of the range of applicability of the perspective neutral framework. Our framework, however, will apply, and we expect to see similar decoherence effects to those reported in the quantum information literature. We note that there has already been some progress in applying our framework to non-ideal frames, in the master thesis of Sébastien Garmier, supervised by Esteban Castro <https://www.research-collection.ethz.ch/handle/20.500.11850/620796>.

Changes to the manuscript: We have added, in the third paragraph of page 5, a comment to emphasise that the zero-charge constraint in Dirac quantisation stems from a specific Lagrangian and not from symmetry considerations alone. We have also mentioned <https://arxiv.org/abs/2302.06603> in the discussion, as an example in quantum gravity where one does not impose a total charge equal to zero in the Hamiltonian constraint. We have also included a citation to the application of our work to non-ideal reference frames.

Minor comment:

In the second to last paragraph of p.5 it is written "...the jumping rules of Refs. [37, 39] could be valid for any reference frames A and B only in two circumstances:..." but then only circumstance 1) is mentioned. Is it a typo or am I missing something

AR:

We thank the reviewer for this comment. This part was rewritten and replaced by points 1) and 2) above in the same paragraph. We now fixed it by deleting the sentence.

We thank the reviewer for their remarks and for their recommendation to accept our paper despite the remaining disagreement. We respond to the reviewer's remarks below.

I thank the authors for their detailed reply and the numerous clarifications added to the manuscript. However, I respectfully disagree with two points in the authors's reply.

The first is the statement that the imposition of the constraints in the sense of Dirac quantization, and hence the need of a coherent group averaging, would not be required by symmetry considerations alone and would rather require to further impose a charge sector.

The second strictly related point is again about external frame independence. Let consider for example a system of N particles in 1 dimension. We can think of the absolute positions of the particles (the kinematical degrees of freedom) as being specified relative to the position of some other particle external to the system playing the role of an external frame and whose position is set to be say in the origin. Now, wiping out any reference to the external frame amounts to require the total momentum of the particles to be vanishing.

We respectfully disagree with the Reviewer on this point. We would like to note that we have already responded to this critique in the second paragraph of page 10 of the previous version of our manuscript, noting that "if an observer loses access to the reference frame relative to which their description of the system is given, they would still be able to make sense of the subset of relative operators that are localized entirely on the system, and these are exactly the set of invariant operators of the system". If the symmetry group in question is the translation group, as the reviewer suggests above, then these degrees of freedom include the total momentum. Hence, there is no reason why we should expect it to vanish based only on symmetry grounds. The situation is similar for other charges, depending on the relevant group. In the case of Dirac quantisation that leads to a constraint on the total momentum, the symmetry is in fact stronger. Furthermore, we note that the total momentum need not vanish for the relevant type of symmetry even under Dirac quantisation. To show explicitly that coherent group averaging is not implied by requiring a gauge symmetry alone, in Methods: Lagrangians for translation gauge symmetry we have constructed two Lagrangians, each of which is invariant, up to a total derivative, under the same gauge transformations $q \rightarrow q + f(t)$. We show that each Lagrangian leads to a different constraint on the total charge.

Gauge transformations, which amount to a translation of the N particles, can in fact be equivalently thought of as a translation of the external frame and gauge-invariant states are those annihilated by the constraint. This does not necessarily require to invoke arguments about a Lagrangian and dynamics. Clearly, it would also be true if one starts with a translation-invariant Lagrangian but the above considerations holds only on the base of requiring gauge-invariance/external frame-independence. A purely internal description of the system of particles then requires not only the observables to be gauge-invariant but also the states to satisfy the constraints. The gauge-invariant states are obtained via coherent group averaging and, upon

completion with respect to a suitable inner product, form the physical Hilbert space of Dirac quantization. For this reason, I think that a superposition and mixture of gauge-related states should not be distinguishable without having access to the external frame (and this is not based on classical intuition). I therefore think of the fact that these states can be distinguished by operations in the invariant algebra considered by the authors as signalling that some remnant of the external frame information is still present if one only requires invariance of observables under unitary conjugation without considering what is happening at the level of the states. The considerations in Appendix A are not in contrast with this as one can easily convince himself that, for any kinematical operator A , the following relation holds true $\Pi A \Pi = G(A)\Pi$, where Π denotes the coherent group averaging projector and $G(A)$ the G -twirl, i.e., coherent and incoherent group averaging agree on the physical Hilbert space. A fully external frame-independent description should however be such both at the level of observables and states and is achieved on H_{phys} .

The rest of the argument by the referee relies strongly on the assumption of vanishing momentum, against which we have presented compelling arguments already.

According to our framework, the $P = 0$ condition found in the perspective neutral framework could only be correct in two cases: 1) P is not physical. One cannot measure it and fixing $P=0$ is a convention. But in this case the internal momenta used in that framework are not defined relative to internal frames for momentum or velocity; they are just postulated conjugate quantities that need to be redefined every time a new system is added. In the section previously called "The problem", which is now part of the introduction, we point out the difficulties that come, in this case, when bringing in new systems under consideration. As we point out, overcoming these problems is one of the motivations for our work. 2) We have translation symmetry, in the sense that we lack a reference frame for the translation group. In this case, $P=0$ is given by the implicit use of a reference frame for momentum, relative to which P is measured and found to vanish. The predictions in both cases agree, but the meanings of the quantities are very different. A closer look at our analysis of the Galilei group clarifies the connection between these two pictures. As explained in the last paragraph of QRFs for the centrally extended Galilei group, our QRF for Galilei can be understood in terms of a reference frame for position ($A_{\{m_1\}}$) and a reference frame for velocity ($A_{\{m_2\}}$). The translation group is obtained from the Galilei group by tacitly assuming access to $A_{\{m_2\}}$ and focusing only on $A_{\{m_1\}}$. For translations, the $P = 0$ condition is equivalent to requiring that the extra particle is in the state $|P=0\rangle$. Now, eq. 32b expresses the fact that the momentum of the extra particle is nothing but the relative momentum of the reference frame for position, $A_{\{m_1\}}$, and the system, S , with respect to the reference frame for momentum, $A_{\{m_2\}}$. This reveals that the condition $P=0$ for translations is nothing else than requiring that the momentum of the extra particle vanishes, which in turn means

that the momentum of the system and the reference frame for translations, relative to the reference frame for velocity, should be zero.

A similar version of this argument is formulated in the last paragraph of page 24 and in the following paragraph of page 25. In the revised version of our manuscript, we have linked this argument to eq. 32b, on which we have expanded slightly.

In conclusion, we believe that we have provided strong evidence in favour of our framework and interpretation of quantum reference frame transformations. In view of this, we would like to invite the Reviewer to reconsider their standpoint on the matter.

This being said, I think the paper represents an interesting contribution to the QRF literature and the authors addressed most of my previous comments and clarified their motivations to a sufficient degree to make the manuscript reasonable to publish. I would nevertheless appreciate to hear the authors' opinion on the above discussion and, in case they agree, modify/adjust accordingly the comments added on page 5 and 10 of the revised manuscript.

We would like to thank the Reviewer for their interesting comments and their overall support. Their remarks have definitely helped us sharpen our arguments and improve the quality of our paper.